# C-MinHash: Improving Minwise Hashing with Circulant Permutation

## Abstract

Minwise hashing (MinHash) is an important and practical algorithm for generating random hashes to approximate the Jaccard (resemblance) similarity in massive binary (0/1) data. The basic theory of MinHash requires applying hundreds or even thousands of independent random permutations to each data vector in the dataset, in order to obtain reliable results for (e.g.,) building large-scale learning models or approximate near neighbor search in massive data. In this paper, we propose **Circulant MinHash (C-MinHash)** and provide the surprising theoretical results that using only **two** independent random permutations in a circulant manner leads to uniformly smaller Jaccard estimation variance than that of the classical MinHash with $K$ independent permutations. Experiments are conducted to show the effectiveness of the proposed method. We also analyze a more convenient C-MinHash variant which reduces two permutations to just one, with extensive numerical results to validate that it achieves essentially the same estimation accuracy as using two permutations with rigorous theory.

## 1 Introduction

Given two $D$-dimensional binary vectors $\boldsymbol{v}, \boldsymbol{w} \in \{0,1\}^D$, the Jaccard similarity is defined as

$$J(\boldsymbol{v}, \boldsymbol{w}) = \frac{\sum_{i=1}^{D} \mathbb{1}\{\boldsymbol{v}_i = \boldsymbol{w}_i = 1\}}{\sum_{i=1}^{D} \mathbb{1}\{\boldsymbol{v}_i + \boldsymbol{w}_i \geq 1\}}, \tag{1}$$

which is a commonly used similarity metric in machine learning and web search applications. The vectors $\boldsymbol{v}$ and $\boldsymbol{w}$ can also be viewed as two sets of items (which represent the locations of non-zero entries), the Jaccard similarity can be equivalently viewed as the size of set intersection over the size of set union. The well-known method of *"minwise hashing"* (MinHash) (Broder, 1997; Broder et al., 1997; 1998; Li & Church, 2005; Li & König, 2011) is a standard technique for computing/estimating the Jaccard similarity in massive binary datasets, with numerous applications such as near neighbor search, duplicate detection, malware detection, web search, clustering, large-scale learning, social networks, computer vision, etc. (Charikar, 2002; Fetterly et al., 2003; Henzinger, 2006; Das et al., 2007; Buehrer & Chellapilla, 2008; Bendersky & Croft, 2009; Chierichetti et al., 2009; Lee et al., 2010; Li et al., 2011; Deng et al., 2012; Chum & Matas, 2012; Shrivastava & Li, 2012; He et al., 2013; Tamersoy et al., 2014; Shrivastava & Li, 2014; Zamora et al., 2016).

### 1.1 A Review of Minwise Hashing (MinHash)

---

**Algorithm 1** Minwise-hashing (MinHash)

---

**Input:** Binary data vector $\boldsymbol{v} \in \{0,1\}^D$,     $K$ independent permutations $\pi_1, ..., \pi_K: [D] \to [D]$.

**Output:** $K$ hash values $h_1(\boldsymbol{v}), ..., h_K(\boldsymbol{v})$.

For $k = 1$ to $K$

   $h_k(\boldsymbol{v}) \leftarrow \min_{i:v_i \neq 0} \pi_k(i)$

End For

---

For simplicity, Algorithm 1 considers just one vector $\boldsymbol{v} \in \{0, 1\}^D$. In order to generate $K$ hash values for $\boldsymbol{v}$, we assume $K$ independent permutations: $\pi_1, ..., \pi_K : [D] \mapsto [D]$. For each permutation, the hash value is the first non-zero location in the permuted vector, i.e., $h_k(\boldsymbol{v}) = \min_{i:v_i \neq 0} \pi_k(i)$, $\forall k = 1, ..., K$. Similarly, for another binary vector $\boldsymbol{w} \in \{0, 1\}^D$, using the same $K$ permutations, we can also obtain $K$ hash values, $h_k(\boldsymbol{w})$. The estimator of $J(\boldsymbol{v}, \boldsymbol{w})$ is simply

$$\hat{J}_{MH}(\boldsymbol{v}, \boldsymbol{w}) = \frac{1}{K} \sum_{k=1}^{K} \mathbb{1}\{h_k(\boldsymbol{v}) = h_k(\boldsymbol{w})\}, \tag{2}$$

where $\mathbb{1}\{\cdot\}$ is the indicator function. By fundamental probability and the independence among the permutations, it is easy to show that

$$\mathbb{E}[\hat{J}_{MH}] = J, \qquad Var[\hat{J}_{MH}] = \frac{J(1 - J)}{K}. \tag{3}$$

How large is $K$? The answer depends on the application domains. For example, for training large-scale machine learning models, it appears that $K = 512$ or $K = 1024$ might be sufficient (Li et al., 2011). However, for approximate near neighbor search using many hash tables (Indyk & Motwani, 1998), it is likely that $K$ might have to be much larger than 1024 (Shrivastava & Li, 2012; 2014).

In the early work of MinHash (Broder, 1997; Broder et al., 1997), actually only one permutation was used by storing the first $K$ non-zero locations after the permutation. Later Li & Church (2005) proposed better estimators to improve the estimation accuracy. The major drawback of the original scheme was that the hashed values did not form a metric space (e.g., satisfying the triangle inequality) and hence could not be used in many algorithms/applications. We believe this was the main reason why the original authors moved to using $K$ permutations (Broder et al., 1998).

## 1.2 OUTLINE OF MAIN RESULTS

**From $K$ Permutations to two.** The main idea of this work, is to replace the independent permutations in MinHash with "circulant" permutations. Thus, we name the proposed framework **C-MinHash** (circulant MinHash). The "circulant" trick was used in the literature of random projections. For example, Yu et al. (2017) showed that using circulant projections hurts the estimation accuracy, but not by too much when the data are sparse. In Section 3, we present some (perhaps surprising) theoretical findings that we just need 2 permutations in MinHash and the results (estimation variances) are even more accurate. Basically, with the **initial permutation** (denoted by $\sigma$), we randomly shuffle the data to break whatever structure which might exist in the original data, and then the **second permutation** (denoted by $\pi$) is applied and re-used $K$ times to generate $K$ hash values, via circulation. This method is called C-MinHash-$(\sigma, \pi)$. Before that, in Section 2, we analyze a simpler variant C-MinHash-$(0, \pi)$ without initial permutation $\sigma$. Although it is not our recommended method, our analysis for C-MinHash-$(0, \pi)$ provides the necessary preparation for later methods and the intuition to understand the need for the initial permutation.

**From Two Permutations to one.** In Section 5, we provide a more convenient variant, C-MinHash-$(\pi, \pi)$, that only needs one permutation $\pi$ for both pre-processing and hashing. Although the theoretical analysis becomes very complicated, we are able to explicitly write down the expectation of the estimator, which is no longer unbiased but the bias is extremely small and has essentially no impact on the estimation accuracy (mean square errors). Extensive experiments are provided to verify that this variant has same estimation accuracy as C-MinHash-$(\sigma, \pi)$ using two permutations.

In this paper, we mainly focus on presenting the fundamental idea of C-MinHash and the theoretical analysis on its uniform variance reduction with non-trivial efforts. Besides improving the Jaccard estimation accuracy when directly implemented in place of MinHash, the proposed C-MinHash mechanism can also be conveniently used as a tool to improve more MinHash based algorithms, for example, the One Permutation Hashing (OPH, Li et al. (2012)). When combined with OPH, C-MinHash can reduce the required one permutation to effectively $1/K$ permutation only. Since such combination is a research direction which requires independent introduction and efforts, we present the work in a separate manuscript (which could be anonymously shared with Referees if needed).

---

**Algorithm 2** C-MinHash-$(0, \pi)$

---

**Input:** Binary data vector $\boldsymbol{v} \in \{0, 1\}^D$,     Permutation vector $\pi$: $[D] \to [D]$

**Output:** Hash values $h_1(\boldsymbol{v}), ..., h_K(\boldsymbol{v})$

For $k = 1$ to $K$

    Shift $\pi$ circularly rightwards by $k$ units: $\pi_k = \pi_{\to k}$

    $h_k(\boldsymbol{v}) \leftarrow \min_{i: v_i \neq 0} \pi_{\to k}(i)$

End For

---

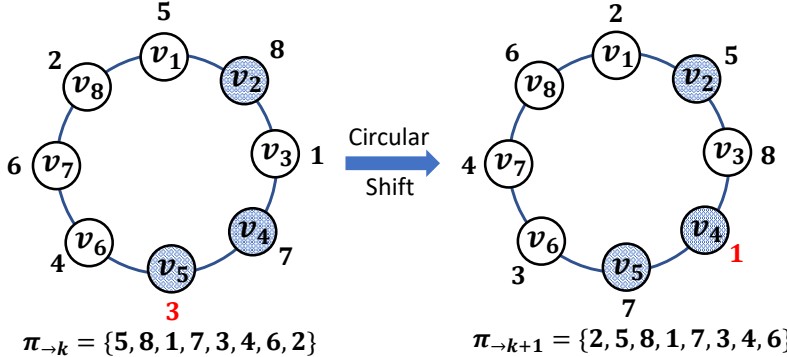

Figure 1: An illustration of the idea of C-MinHash. The data vector has three non-zeros, $v_2 = v_4 = v_5 = 1$. In this example, $h_k(\boldsymbol{v}) = 3$, $h_{k+1}(\boldsymbol{v}) = 1$.

## 2   C-MINHASH-$(0, \pi)$ WITHOUT INITIAL PERMUTATION

As shown in Algorithm 2, the C-MinHash algorithm has similar operations as MinHash. The difference lies in the permutations used in the hashing process. To generate each hash $h_k(\boldsymbol{v})$, we permute the data vector using $\pi_{\to k}$, which is the permutation shifted $k$ units circularly towards right based on $\pi$. For example, $\pi = [3, 1, 2, 4], \pi_{\to 1} = [4, 3, 1, 2], \pi_{\to 2} = [2, 4, 3, 1]$, etc. Conceptually, we may think of circulation as concatenating the first and last elements of a vector to form a circle; see Figure 1. We set the hash value $h_k(\boldsymbol{v})$ as the position of the first non-zero after being permuted by $\pi_{\to k}$. Analogously, we define the C-MinHash-$(0, \pi)$ estimator of the Jaccard similarity $J(\boldsymbol{v}, \boldsymbol{w})$ as

$$\hat{J}_{0,\pi} = \frac{1}{K} \sum_{k=1}^{K} \mathbb{1}\{h_k(\boldsymbol{v}) = h_k(\boldsymbol{w})\}, \tag{4}$$

where $h$ is the hash value output by Algorithm 2. In this paper, for simplicity, we assume $K \leq D$.

Next, we present the theoretical analysis for Algorithm 2, in terms of the expectation (mean) and the variance of the estimator $\hat{J}_{0,\pi}$. Our results reveal that the estimation accuracy depends on the initial data distribution, which may lead to undesirable performance behaviors when real-world datasets exhibit various structures. On the other hand, while it is not our recommended method, the analysis serves a preparation (and insight) for the C-MinHash-$(\sigma, \pi)$ which will soon be described.

First, we introduce some notations and definitions. Given $\boldsymbol{v}, \boldsymbol{w} \in \{0, 1\}^D$, we define $a$ and $f$ as

$$a = \sum_{i=1}^{D} \mathbb{1}\{\boldsymbol{v}_i = 1 \text{ and } \boldsymbol{w}_i = 1\}, \quad f = \sum_{i=1}^{D} \mathbb{1}\{\boldsymbol{v}_i = 1 \text{ or } \boldsymbol{w}_i = 1\}. \tag{5}$$

We say that $(\boldsymbol{v}, \boldsymbol{w})$ is a $(D, f, a)$-*data pair*, whose Jaccard similarity can also be written as $J = a/f$.

**Definition 2.1.** *Consider $\boldsymbol{v}, \boldsymbol{w} \in \{0, 1\}^D$. Define the **location vector** as $\boldsymbol{x} \in \{O, \times, -\}^D$, with $\boldsymbol{x}_i$ being "$O$", "$\times$", "$-$", when $\boldsymbol{v}_i = \boldsymbol{w}_i = 1$, $\boldsymbol{v}_i + \boldsymbol{w}_i = 1$ and $\boldsymbol{v}_i = \boldsymbol{w}_i = 0$, respectively.*

The location vector $\boldsymbol{x}$ can fully characterize a hash collision. When a permutation $\pi_{\to k}$ is applied, the hashes $h_k(\boldsymbol{v})$ and $h_k(\boldsymbol{w})$ would collide if after permutation, the first "$O$" is placed before the first "$\times$" (counting from small to large). This observation will be the key in our theoretical analysis.

**Definition 2.2.** *For $A, B \in \{O, \times, -\}$, let $\{(A, B)|\triangle\}$ denote the set $\{(i, j) : (\boldsymbol{x}_i, \boldsymbol{x}_j) = (A, B), j - i = \triangle\}$. For each $1 \leq \triangle \leq K - 1$, define*

$$\mathcal{L}_0(\triangle) = \{(O, O)|\triangle\}, \ \mathcal{L}_1(\triangle) = \{(O, \times)\}, \ \mathcal{L}_2(\triangle) = \{(O, -)\},$$
$$\mathcal{G}_0(\triangle) = \{(-, O)|\triangle\}, \ \mathcal{G}_1(\triangle) = \{(-, \times)\}, \mathcal{G}_2(\triangle) = \{(-, -)\},$$
$$\mathcal{H}_0(\triangle) = \{(\times, O)|\triangle\}, \ \mathcal{H}_1(\triangle) = \{(\times, \times)\}, \ \mathcal{H}_2(\triangle) = \{(\times, -)\}.$$

**Remark 2.1.** *For the ease of notation, by circulation we write $\boldsymbol{x}_j = \boldsymbol{x}_{j-D}$ when $D < j < 2D$.*

Definition 2.2 measures the relative location of different types of points in the location vector, for a specific pair of data vectors. One can easily verify that for $\forall 1 \leq \triangle \leq K - 1$,

$$|\mathcal{L}_0| + |\mathcal{L}_1| + |\mathcal{L}_2| = |\mathcal{L}_0| + |\mathcal{G}_0| + |\mathcal{H}_0| = a,$$
$$|\mathcal{G}_0| + |\mathcal{G}_1| + |\mathcal{G}_2| = |\mathcal{L}_2| + |\mathcal{G}_2| + |\mathcal{H}_2| = D - f, \tag{6}$$
$$|\mathcal{H}_0| + |\mathcal{H}_1| + |\mathcal{H}_2| = |\mathcal{L}_1| + |\mathcal{G}_1| + |\mathcal{H}_1| = f - a,$$

which is the intrinsic constraints on the size of above sets. We are now ready to analyze the expectation and variance of $\hat{J}_{0,\pi}$. It is easy to see that $\hat{J}_{0,\pi}$ is still unbiased, i.e., $\mathbb{E}[\hat{J}_{0,\pi}] = J$, by linearity of expectation. Lemma 2.1 provides an important quantity that leads to $Var[\hat{J}_{0,\pi}]$ as in Theorem 2.2.

**Lemma 2.1.** *For any $1 \leq s < t \leq K$ with $t - s = \triangle$, we have*

$$\mathbb{E}_\pi\big[\mathbb{1}\{h_s(\boldsymbol{v}) = h_s(\boldsymbol{w})\}\mathbb{1}\{h_t(\boldsymbol{v}) = h_t(\boldsymbol{w})\}\big] = \frac{|\mathcal{L}_0(\triangle)| + (|\mathcal{G}_0(\triangle)| + |\mathcal{L}_2(\triangle)|)J}{f + |\mathcal{G}_0(\triangle)| + |\mathcal{G}_1(\triangle)|},$$

*where the sets are defined in Definition 2.2 and $h_s$, $h_t$ are the hash values as in Algorithm 2.*

**Theorem 2.2.** *Under the same setting as in Lemma 2.1, the variance of $\hat{J}_{0,\pi}$ is*

$$Var[\hat{J}_{0,\pi}] = \frac{J}{K} + \frac{2\sum_{s=2}^K (s-1)\Theta_{K-s+1}}{K^2} - J^2,$$

*where $\Theta_\triangle \triangleq E_\pi\big[\mathbb{1}\{h_s(\boldsymbol{v}) = h_s(\boldsymbol{w})\}\mathbb{1}\{h_t(\boldsymbol{v}) = h_t(\boldsymbol{w})\}\big]$ as in Lemma 2.1 with any $t - s = \triangle$.*

From Theorem 2.2, we see that the variance of $\hat{J}_{0,\pi}$ depends on $a$, $f$, and the size of sets $\mathcal{L}$'s and $\mathcal{G}$'s as in Definition 2.1, which is determined by the location vector $\boldsymbol{x}$. Since we use the original data vectors without randomly permuting the entries beforehand, $Var[\hat{J}_{0,\pi}]$ is called "*location-dependent*" as it is dependent on the location of non-zero entries of the original data.

## 3 C-MINHASH-$(\sigma, \pi)$ WITH INDEPENDENT INITIAL PERMUTATION

---

**Algorithm 3** C-MinHash-$(\sigma, \pi)$

---

**Input:** Binary data vector $\boldsymbol{v} \in \{0, 1\}^D$,      Permutation vectors $\pi$ and $\sigma$: $[D] \to [D]$

**Output:** Hash values $h_1(\boldsymbol{v}), ..., h_K(\boldsymbol{v})$

Initial permutation: $\boldsymbol{v}' = \sigma(\boldsymbol{v})$

For $k = 1$ to $K$

     Shift $\pi$ circularly rightwards by $k$ units: $\pi_k = \pi_{\to k}$

     $h_k(\boldsymbol{v}) \leftarrow \min_{i:v'_i \neq 0} \pi_{\to k}(i)$

End For

---

The method C-MinHash-$(\sigma, \pi)$ is summarized in Algorithm 3, which is very similar to Algorithm 2. This time, as pre-processing, we apply an initial permutation $\sigma \perp\!\!\!\perp \pi$ on the data to break whatever structures which might exist. Similarly, we define the C-MinHash-$(\sigma, \pi)$ estimator of $J$ as

$$\hat{J}_{\sigma,\pi} = \frac{1}{K}\sum_{k=1}^K \mathbb{1}\{h_k(\boldsymbol{v}) = h_k(\boldsymbol{w})\}, \tag{7}$$

where $h_k$'s are the hash values output by Algorithm 3. We will present our detailed theoretical analysis and the main result (Theorem 3.5). First, by linearity of expectation and that $\sigma$ and $\pi$ are independent, $\hat{J}_{\sigma,\pi}$ is still an unbiased estimator $\mathbb{E}[\hat{J}_{\sigma,\pi}] = \mathbb{E}[\mathbb{1}\{h_i(\boldsymbol{v}) = h_i(\boldsymbol{w})\}] = J$. To provide an immediate intuition, the following proposition emphasizes that the collision indicator variables in (7) are *pairwise negatively correlated*, which intuitively explains the source of variance reduction.

**Proposition 3.1.** *In* (7), $Cov(\mathbb{1}\{h_i(\boldsymbol{v}) = h_i(\boldsymbol{w})\}, \mathbb{1}\{h_j(\boldsymbol{v}) = h_j(\boldsymbol{w})\}) < 0, \forall i \neq j.$

*Proof.* Denote $\tilde{\mathcal{E}} = \mathbb{E}_{i \neq j}[\mathbb{1}\{h_i(\boldsymbol{v}) = h_i(\boldsymbol{w})\}\mathbb{1}\{h_j(\boldsymbol{v}) = h_j(\boldsymbol{w})\}]$, for $i \neq j$. The task is then to show that $Cov(\mathbb{1}\{h_i(\boldsymbol{v}) = h_i(\boldsymbol{w})\}, \mathbb{1}\{h_j(\boldsymbol{v}) = h_j(\boldsymbol{w})\}) = \tilde{\mathcal{E}} - J^2 < 0$. By Theorem 3.5, $Var[\hat{J}_{MH}] < Var[\hat{J}_{\sigma,\pi}]$, where $Var[\hat{J}_{MH}] = \frac{J(1-J)}{K}$ and

$$Var[\hat{J}_{\sigma,\pi}] = \mathbb{E}\left[\left(\hat{J}_{\sigma,\pi}\right)^2\right] - J^2 = \mathbb{E}\left[\left(\frac{1}{K}\sum_{k=1}^{K}\mathbb{1}\{h_k(\boldsymbol{v}) = h_k(\boldsymbol{w})\}\right)^2\right] - J^2$$

$$= \frac{\sum_i \mathbb{E}[\mathbb{1}\{h_i(\boldsymbol{v}) = h_i(\boldsymbol{w})\}]}{K^2} + \frac{\sum_{i \neq j}\mathbb{E}[\mathbb{1}\{h_i(\boldsymbol{v}) = h_i(\boldsymbol{w})\}\mathbb{1}\{h_j(\boldsymbol{v}) = h_j(\boldsymbol{w})\}]}{K^2} - J^2$$

$$= \frac{J}{K} + \frac{K(K-1)\mathbb{E}[\mathbb{1}\{h_i(\boldsymbol{v}) = h_i(\boldsymbol{w})\}\mathbb{1}\{h_j(\boldsymbol{v}) = h_j(\boldsymbol{w})\}]}{K^2} - J^2, \quad \text{where } i \neq j$$

$$= \frac{J}{K} + \frac{(K-1)\tilde{\mathcal{E}}}{K} - J^2 < \frac{J(1-J)}{K} \implies \left(\tilde{\mathcal{E}} - J^2\right)\left(1 - \frac{1}{K}\right) < 0 \implies \tilde{\mathcal{E}} - J^2 < 0. \qquad \square$$

Next, we formally present the path to our main result. The next Theorem gives the variance of $\hat{J}_{\sigma,\pi}$.

**Theorem 3.2.** *Let $a, f$ be defined as in* (5). *When $0 < a < f \leq D$ ($J \notin \{0,1\}$), we have*

$$Var[\hat{J}_{\sigma,\pi}] = \frac{J}{K} + \frac{(K-1)\tilde{\mathcal{E}}}{K} - J^2, \tag{8}$$

*where with $l = \max(0, D - 2f + a)$, and*

$$\tilde{\mathcal{E}} = \mathbb{E}_{i \neq j}[\mathbb{1}\{h_i(\boldsymbol{v}) = h_i(\boldsymbol{w})\}\mathbb{1}\{h_j(\boldsymbol{v}) = h_j(\boldsymbol{w})\}] = \sum_{\{l_0, l_2, g_0, g_1\}}\left\{\left(\frac{l_0}{f + g_0 + g_1} + \frac{a(g_0 + l_2)}{(f + g_0 + g_1)f}\right)\right.$$

$$\left. \times \sum_{s=l}^{D-f-1}\frac{\binom{D-f}{s}}{\binom{D-a-1}{D-f-1}}\frac{\binom{f-a-1}{D-f-s-1}\binom{s}{n_1}\binom{D-f-s}{n_2}\binom{D-f-s}{n_3}\binom{f-a-(D-f-s)}{n_4}\binom{a-1}{a-l_1-l_2}}{\binom{D-1}{a}}\right\}. \tag{9}$$

*The feasible set $\{l_0, l_2, g_0, g_1\}$ satisfies the intrinsic constraints* (6), *and*

$$n_1 = g_0 - (D - f - s - g_1), \qquad n_2 = D - f - s - g_1,$$
$$n_3 = l_2 - g_0 + (D - f - s - g_1), \quad n_4 = l_1 - (D - f - s - g_1).$$

*Also, when $a = 0$ or $f = a$ ($J = 0$ or $J = 1$), we have $Var[\hat{J}_{\sigma,\pi}] = 0$.*

Since the original locational structure of the data is broken by the initial permutation $\sigma$, $Var[\hat{J}_{\sigma,\pi}]$ only depends on the value of $(D, f, a)$, i.e., it is *"location-independent"*. This would make the performance of C-MinHash-$(\sigma, \pi)$ consistent in different tasks. Same as MinHash, Proposition 3.3 states that given $D$ and $f$, the variance of $\hat{J}_{\sigma,\pi}$ is symmetric about $J = 0.5$, as illustrated in Figure 2, which also shows that the variance of $\hat{J}_{\sigma,\pi}$ is smaller than the variance of the original MinHash.

**Proposition 3.3** (Symmetry). *$Var[\hat{J}_{\sigma,\pi}]$ is the same for the $(D, f, a)$-data pair and the $(D, f, f - a)$-data pair, $\forall 0 \leq a \leq f \leq D$.*

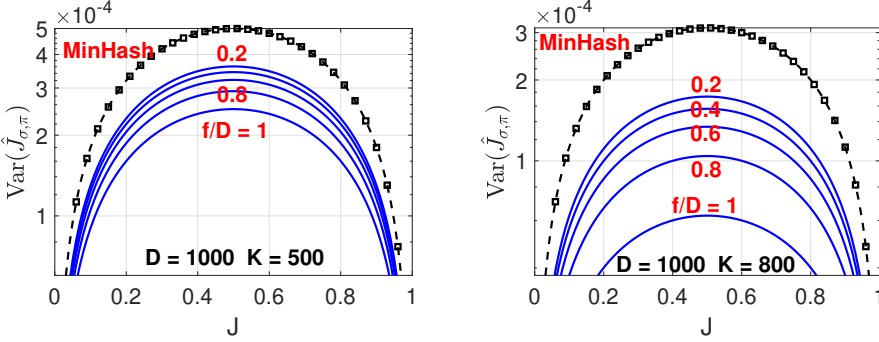

Figure 2: $Var[\hat{J}_{\sigma,\pi}]$ versus $J$, with $D = 1000$ and varying $f$. **Left:** $K = 500$. **Right:** $K = 800$.

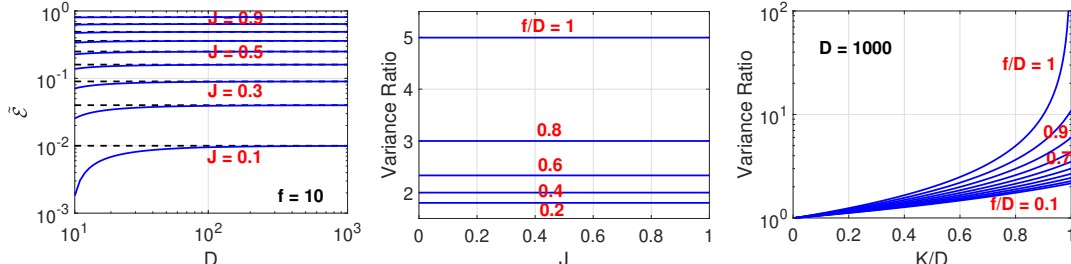

Figure 3: **Left:** Theoretical $\tilde{\mathcal{E}}$, $f = 10$. Each dash line represents the corresponding $J^2$. **Mid:** Variance ratio $\frac{Var[\hat{J}_{MH}]}{Var[\hat{J}_{\sigma,\pi}]}$, $D = 1000$, $K = 800$. **Right:** Variance ratio $\frac{Var[\hat{J}_{MH}]}{Var[\hat{J}_{\sigma,\pi}]}$, $D = 1000$.

A rigorous comparison of $Var[\hat{J}_{\sigma,\pi}]$ and $Var[\hat{J}_{MH}]$ appears to be a challenging task given the complicated combinatorial form of $Var[\hat{J}_{\sigma,\pi}]$. The following lemma characterizes an important property of $\tilde{\mathcal{E}}$ in (9), that it is monotone in $D$ when $a$ and $f$ are fixed, as illustrated in Figure 3 (left).

**Lemma 3.4** (Increasing Increment). *Assume $f > a > 0$ are arbitrary and fixed. Denote $\tilde{\mathcal{E}}_D$ as in (9) in Theorem 3.2, with $D$ treated as a parameter. Then we have $\tilde{\mathcal{E}}_{D+1} > \tilde{\mathcal{E}}_D$ for $\forall D \geq f$.*

Equipped with Lemma 3.4, we arrive at the following main theoretical result of this work, on the uniform variance reduction of C-MinHash-$(\sigma, \pi)$.

**Theorem 3.5** (Uniform Superiority). *For any two binary vectors $\boldsymbol{v}, \boldsymbol{w} \in \{0, 1\}^D$ with $J \neq 0$ or $1$, it holds that $Var[\hat{J}_{\sigma,\pi}(\boldsymbol{v}, \boldsymbol{w})] < Var[\hat{J}_{MH}(\boldsymbol{v}, \boldsymbol{w})]$.*

That is, using merely two permutations as C-MinHash-$(\sigma, \pi)$ improves the Jaccard estimation variance of classical MinHash, in all cases. Next, in Figure 3 (mid) and Theorem 3.6, we show that interestingly, the relative improvement is actually the same for any $J$, for given $D$, $f$ and $K$.

**Proposition 3.6** (Consistent Improvement). *Suppose $f$ is fixed. In terms of $a$, the variance ratio $\frac{Var[\hat{J}_{MH}(\boldsymbol{v}, \boldsymbol{w})]}{Var[\hat{J}_{\sigma,\pi}(\boldsymbol{v}, \boldsymbol{w})]}$ is constant for any $0 < a < f$.*

How is the improvement affected by the sparsity (i.e., $f$) and the number of hashes $K$? In Figure 3 (right), we plot the variance ratio $\frac{Var[\hat{J}_{MH}]}{Var[\hat{J}_{\sigma,\pi}]}$ with different $f$ and $K$, given fixed $D$. We see that the improvement in variance increases with $K$ (more hashes) and $f$ (more non-zero entries). Note that, by Proposition 3.6, here we do not need to consider $a$ which does not affect the variance ratio. The results in Figure 3 again verify Theorem 3.5, as the variance ratio is always greater than 1.

## 4 NUMERICAL EXPERIMENTS

In this section, we provide numerical experiments to validate our theoretical findings and demonstrate that C-MinHash can indeed lead to smaller Jaccard estimation errors.

### 4.1 SANITY CHECK: A SIMULATION STUDY

A simulation study is conducted on synthetic data to verify the theoretical variances given by Theorem 2.2 and Theorem 3.2. We simulate $D = 128$ dimensional binary vector pairs $(\boldsymbol{v}, \boldsymbol{w})$ with different $f$ and $a$, which have a special locational structure that the location vector $\boldsymbol{x}$ is such that $a$ "$O$"'s are followed by $(f - a)$ "$\times$"'s and then followed by $(D - f)$ "$-$"'s sequentially. We plot the empirical and theoretical mean square errors (MSE = variance + bias$^2$) in Figure 4:

- The theoretical variance agrees with the empirical observation, as the curves overlap, confirming Theorem 2.2 and Theorem 3.2. The variance reduction increases with larger $K$.

- $Var[\hat{J}_{\sigma,\pi}]$ (C-MinHash-$(\sigma, \pi)$) is always smaller than $Var[\hat{J}_{MH}]$, as stated by Theorem 3.5. In contrast, $Var[\hat{J}_{0,\pi}]$ (C-MinHash-$(0, \pi)$) varies significantly depending on different data structures, as mentioned in Section 2.

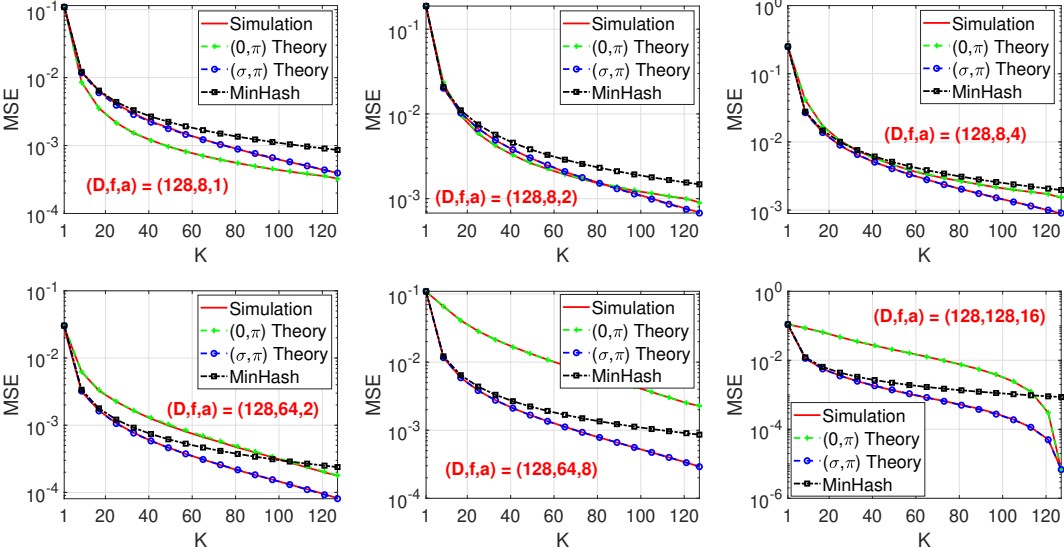

Figure 4: Empirical vs. theoretical variance of $\hat{J}_{0,\pi}$ (C-MinHash-$(0,\pi)$) and $\hat{J}_{\sigma,\pi}$ (C-MinHash-$(\sigma,\pi)$), on synthetic binary data vector pairs with different data patterns.

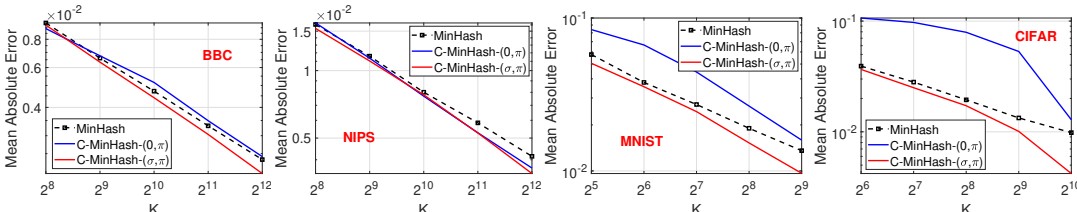

Figure 5: Mean Absolute Error ( MAE) of MinHash and C-MinHash on real-world datasets.

## 4.2 Jaccard Estimation on Text and Image Datasets

We test C-MinHash on four datasets, including two text datasets: the NIPS full paper dataset from UCI repository (Dua & Graff, 2017), and the BBC News dataset (Greene & Cunningham, 2006), and two popular image datasets: the MNIST dataset (LeCun et al., 1998) with hand-written digits, and the CIFAR dataset (Krizhevsky, 2009) containing natural images. All the datasets are processed to be binary. For each dataset with $n$ data vectors, there are in total $n(n-1)/2$ data vector pairs. We estimate the Jaccard similarities for all the pairs and report the mean absolute errors (MAE). The results are averaged over 10 independent repetitions, for each dataset, as shown in Figure 5:

- The MAE of C-MinHash-$(\sigma,\pi)$ is consistently smaller than that of MinHash, again confirming Theorem 3.5 on the benefit of the improved variance. The improvements become more substantial with larger $K$, which is consistent with the trend in Figure 3.

- Without the initial permutation $\sigma$, the accuracy of C-MinHash-$(0,\pi)$ is affected by the distribution of the original data, and it is worse than C-MinHash-$(\sigma,\pi)$ on all these four datasets. Also, the performance of C-MinHash-$(0,\pi)$ on image data seems much worse than that on text data, which we believe is because the image datasets contain more structural patterns. This again suggests that the initial permutation $\sigma$ might be needed in practice.

In summary, the simulation study has verified the correctness of our theoretical findings in Theorem 2.2 and Theorem 3.2. The experiments with Jaccard estimation on four datasets confirm that C-MinHash is more accurate than the original MinHash. The initial permutation $\sigma$ is recommended.

## 5 C-MinHash-$(\pi, \pi)$: Practically Reducing to One Permutation

We propose a more convenient variant, C-MinHash-$(\pi,\pi)$, which only requires one permutation. That is, $\pi$ is used for both pre-processing and circulant hashing. The procedure is the same as Algorithm 3, except that the initial permutation $\sigma$ is replaced by $\pi$. Denote the Jaccard estimator $\hat{J}_{\pi,\pi}$.

The complicated dependency between $\pi$ (initial permutation) and $\pi_{\to k}$ (hashing) makes the theoretical analysis challenging. In the following, we provide the mean of each hash collision indicator.

**Theorem 5.1.** *Assume $K \leq D$ and let $a, f$ be defined as (5). The location vector $\boldsymbol{x}$ is defined in Definition 2.1. Denote $\mathcal{B}_1 = \{i : \boldsymbol{x}_i = O\}$, $\mathcal{B}_2 = \{i : \boldsymbol{x}_i = \times\}$ and $\mathcal{B}_3 = \{i : \boldsymbol{x}_i = -\}$. For $a \leq j \leq D$ and $1 \leq k \leq K$, define*

$$\mathcal{A}_-(j) = \{\boldsymbol{x}_i : (i + k - 1 \bmod D) + 1 \leq j\}, \quad \mathcal{A}_+(j) = \{\boldsymbol{x}_i : (i + k - 1 \bmod D) + 1 > j\}.$$

*Let $n_{-,1}(j) = |\{\boldsymbol{x}_i = O : i \in \mathcal{A}_-(j)\}|$ be the number of "O" points in $\mathcal{A}_-(j)$. Analogously let $n_{-,2}(j), n_{-,3}(j)$ be the number of "$\times$" and "$-$" points in $\mathcal{A}_-(j)$, and $n_{+,1}(j), n_{+,2}(j), n_{+,3}(j)$ be the number of "O", "$\times$" and "$-$" points in $\mathcal{A}_+(j)$. Then, for the $k$-th C-MinHash-$(\pi, \pi)$ hash,*

$$\mathbb{E}[\mathbb{1}\{h_k(\boldsymbol{v}) = h_k(\boldsymbol{w})\}] = \sum_{j=1}^D \sum_{Z \in \Theta_j} P_j(Z) \cdot \left\{ \sum_{i=1}^3 \Psi_i(j) + \sum_{i \in \mathcal{B}_1} (1 - \frac{z_{-,1}}{n_{-,1}(i^*)}) \tilde{P}_1 \right\},$$

*where $P_j(Z)$ is the density function of $Z = (z_{-,k}|_1^3, z_{+,k}|_1^3)$ which follows hyper$(D, D - f, n_{-,k}(j)|_1^3, n_{+,k}(j)|_1^3)$, with domain $\Theta_j$. For $q = 1, 2, 3$, denote $\mathbb{1}_q^\# = \mathbb{1}\{j^\# \in \mathcal{B}_q\}$, and*

$$\Psi_q(j) = \sum_{i \in \mathcal{B}_q, j < i^*} \sum_{p=1}^3 \mathbb{1}_p^\# \left(1 - \frac{z_{-,p}}{n_{-,p}(j)} + \mathbb{1}\{q = 3\}(2\frac{z_{-,p}}{n_{-,p}(j)} - 1)\right)(1 - \frac{z_{+,q}}{n_{+,q}(j)}) \tilde{P}_q \bar{J}_q$$

$$+ \sum_{i \in \mathcal{B}_q, j > i^*} \left[\mathbb{1}_q^\#(1 - \frac{z_{-,q}}{n_{-,q}(j)})(1 - \frac{z_{-,q}}{n_{-,q}(j) - 1}) + \sum_{p \neq q}^3 \mathbb{1}_p^\#(1 - \frac{z_{-,q}}{n_{-,q}(j)})(1 - \frac{z_{-,p}}{n_{-,p}(j)})\right] \tilde{P}_q J^*,$$

*where $i^* = (i + k - 1 \bmod D) + 1$, $i^\# = (i - k - 1 \bmod D) + 1$, $\forall i$. Define $J^* = \frac{a - r_1}{f - r_1 - r_2}$, and*

$$\tilde{P}_1 = \tilde{P}_2 = \frac{1}{r_1 + r_2} \frac{\binom{b_0}{r_3}\binom{D - j - b_0}{r_1 + r_2 - 1}}{\binom{D - f}{r_3}\binom{f}{r_1 + r_2}}, \quad \tilde{P}_3 = \frac{1}{r_3} \frac{\binom{b_0}{r_3 - 1}\binom{D - j - b_0}{r_1 + r_2}}{\binom{D - f}{r_3}\binom{f}{r_1 + r_2}},$$

$$\bar{J}_q = \frac{r_1 - \mathbb{1}\{q = 1\}}{D - j - b_0} + (1 - \frac{r_1 + r_2 - \mathbb{1}\{q \neq 3\}}{D - j - b_0}) J^*, \ q = 1, 2, 3,$$

*where $b_0 = \sum_{k=1}^3 z_{+,k}$, $r_1 = a - z_{-,1} - z_{+,1}$, $r_2 = f - a - z_{-,2} - z_{+,2}$ and $r_3 = D - f - z_{-,3} - z_{+,3}$.*

Theorem 5.1 says that $\mathbb{E}[\mathbb{1}\{h_k(\boldsymbol{v}) = h_k(\boldsymbol{w})\}]$ would be different for different $k$. Figure 6 presents numerical examples to validate the theory and demonstrate the magnitude of bias$^2$ (recall MSE = bias$^2$ + variance), where the simulations match perfectly Theorem 5.1. As the bias of $\mathbb{E}[\mathbb{1}\{h_k(\boldsymbol{v}) =$

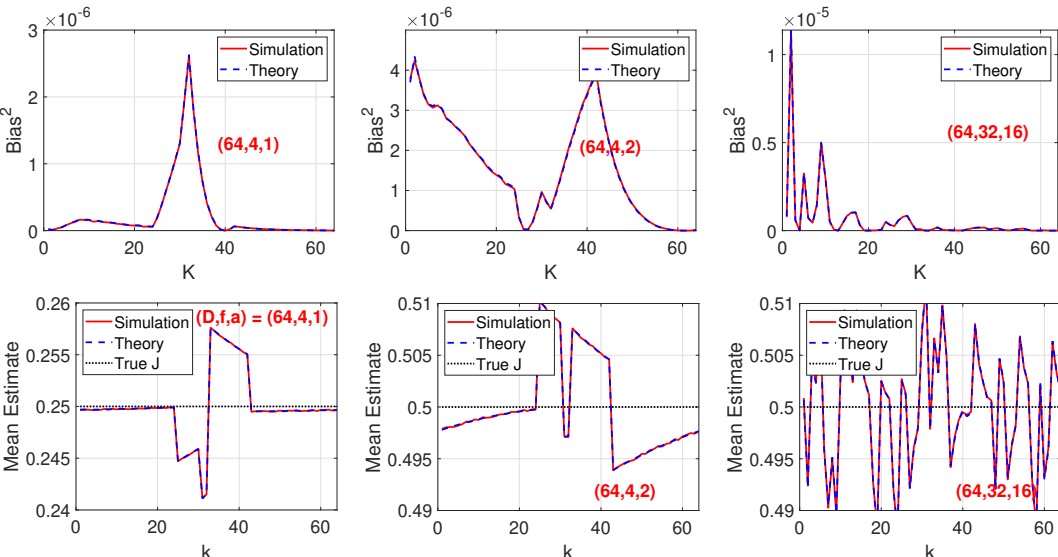

Figure 6: **1st row:** bias$^2 = (\mathbb{E}[\hat{J}_{\pi,\pi}] - J)^2$ vs. number of hashes $K$ on simulated $(D, f, a)$-data pairs, $D = 64$. **2nd row:** mean of each collision indicator.

$h_k(\boldsymbol{w})\}]$ can be positive or negative, the overall bias of $\hat{J}_{\pi,\pi}$ would approach 0 as $K$ increases. From the 2nd row, we see that bias$^2$ is very small ($10^{-5}$ or even smaller) and indeed converges to 0 with more hashes, i.e., the averaging effect. Also, from the proof (see Appendix A.8) we can know that when $a$ and $f$ are fixed, as $D$ increases, $\mathbb{E}[\hat{J}_{\pi,\pi}]$ would converge to $J$.

We found through extensive numerical experiments that, the MSE of $\hat{J}_{\pi,\pi}$ is essentially the same as $\hat{J}_{\sigma,\pi}$. Figure 7 (1st row) compares the empirical of C-MinHash-$(\pi,\pi)$ with the theoretical variances of C-MinHash-$(\sigma,\pi)$, where the simulated data has the same special locational structure as in Section 4.1. In the 2nd row, we present the MAE comparison on real datasets, where we see that the curves for these two estimators ($\hat{J}_{\sigma,\pi}$ and $\hat{J}_{\pi,\pi}$) match well. In all figures, the overlapping MSE curves verify our claim that we just need one permutation $\pi$. Due to the space limitation, we provide more empirical justifications in Appendix B.

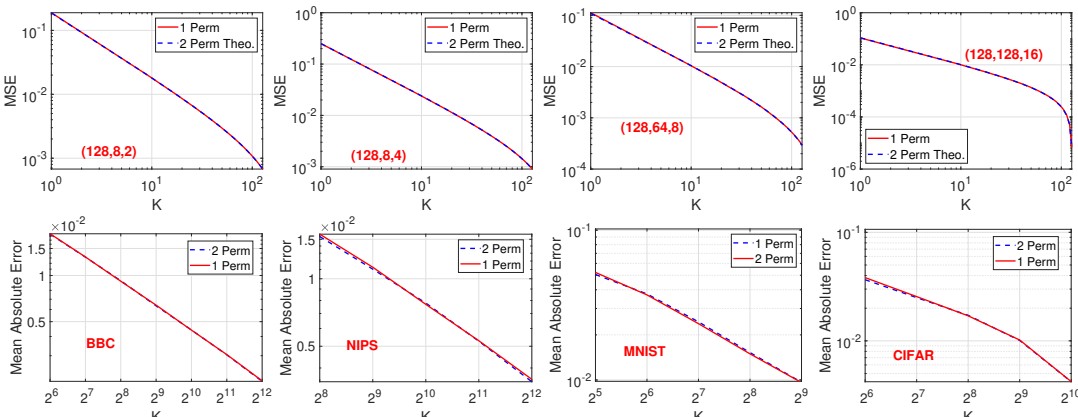

Figure 7: **1st row:** estimator MSE vs. $K$ on simulated data pairs. **2nd row:** MAE of Jaccard estimation on four datasets. "1 Perm" is C-MinHash-$(\pi,\pi)$, and "2 Perm" is C-MinHash-$(\sigma,\pi)$.

## 6 CONCLUSION

The method of *minwise hashing* (MinHash), from the seminal works of Broder and his colleagues, has become standard in industrial practice. One fundamental reason for its wide applicability is that the binary (0/1) high-dimensional representation is convenient and suitable for a wide range of practical scenarios. The so-called Jaccard similarity is a popular measure of similarity in binary data.

To estimate the Jaccard similarity $J$, standard MinHash requires to use $K$ independent permutations, where $K$, the number of hashes, can be several hundreds or even thousands in practice. In this paper, we proposed Circulant MinHash (C-MinHash) present the surprising theoretical results that, with merely 2 permutations, we still obtain an unbiased estimate of the Jaccard similarity with the variance strictly smaller than that of the original MinHash, as confirmed by numerical experiments on simulated and real datasets. The initial permutation is applied to break whatever structure the original data may exhibit. The second permutation is re-used $K$ times in a circulant shifting fashion. Moreover, we analyze a more convenient C-MinHash variance which uses only 1 permutation for both pre-processing and circulant hashing. We derive the complicated mean estimation, and validate through extensive experiments that it has the same Jaccard estimation accuracy as using 2 permutations with strict theoretical guarantee.

Practically speaking, our theoretical results may reveal a useful direction for designing hashing methods. For example, in many applications, using permutation vectors of length (e.g.,) $2^{30}$ might be sufficient. While it is perhaps unrealistic to store (e.g.,) $K = 1024$ such permutation vectors in the memory, one can afford to store two such permutations (even in GPU memory). Using perfectly random permutations in lieu of approximate permutations would simplify the design and analysis of randomized algorithms and ensure that the practical performance strictly matches the theory.

Finally, we mention that our work can be used as a building block to improve Li et al. (2012), reducing the required one permutation to effectively $1/K$ permutation only. If needed, we would be happy to anonymously share the technical manuscript with the related results.

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

# A    PROOFS OF TECHNICAL RESULTS

**Notations.** In our analysis, we will use $\mathbb{1}_s$ to denote $\mathbb{1}\{h_s(\boldsymbol{v}) = h_s(\boldsymbol{w})\}$ in for $\forall 1 \leq s \leq K$, where $h$ is the hash value. Given two data vectors $\boldsymbol{v}, \boldsymbol{w} \in \{0,1\}^D$. Recall in (5): $a = \sum_{i=1}^{D} \mathbb{1}\{\boldsymbol{v}_i = 1$ and $\boldsymbol{w}_i = 1\}$, $f = \sum_{i=1}^{D} \mathbb{1}\{\boldsymbol{v}_i = 1$ or $\boldsymbol{w}_i = 1\}$. Thus, the Jaccard similarity $J = a/f$. We also define $\tilde{J} = (a-1)/(f-1)$.

**Definition A.1.** *Let $\boldsymbol{v}, \boldsymbol{w} \in \{0,1\}^D$. Define the **location vector** as $\boldsymbol{x} \in \{O, \times, -\}^D$, with $\boldsymbol{x}_i$ being "O", "$\times$", "$-$" when $\boldsymbol{v}_i = \boldsymbol{w}_i = 1$, $\boldsymbol{v}_i + \boldsymbol{w}_i = 1$ and $\boldsymbol{v}_i = \boldsymbol{w}_i = 0$, respectively.*

**Definition A.2.** *For $A, B \in \{O, \times, -\}$, let $\{(i,j) : (A,B)|\triangle\}$ denote a pair of indices $\{(i,j) : (\boldsymbol{x}_i, \boldsymbol{x}_j) = (A, B), j - i = \triangle\}$. Define*

$$\mathcal{L}_0(\triangle) = \{(i,j) : (O,O)|\triangle\}, \quad \mathcal{L}_1(\triangle) = \{(i,j) : (O,\times)|\triangle\}, \quad \mathcal{L}_2(\triangle) = \{(i,j) : (O,-)|\triangle\},$$
$$\mathcal{G}_0(\triangle) = \{(i,j) : (-,O)|\triangle\}, \quad \mathcal{G}_1(\triangle) = \{(i,j) : (-,\times)|\triangle\} \ , \mathcal{G}_2(\triangle) = \{(i,j) : (-,-)|\triangle\},$$
$$\mathcal{H}_0(\triangle) = \{(i,j) : (\times,O)|\triangle\}, \quad \mathcal{H}_1(\triangle) = \{(i,j) : (\times,\times)|\triangle\}, \quad \mathcal{H}_2(\triangle) = \{(i,j) : (\times,-)|\triangle\}.$$

**Remark A.1.** *For the ease of notation, by circulation we write $\boldsymbol{x}_j = \boldsymbol{x}_{j-D}$ when $D < j < 2D$.*

One can easily verify that given fixed $a, f, D$, it holds that for $\forall 1 \leq \triangle \leq K - 1$,

$$|\mathcal{L}_0(\triangle)| + |\mathcal{L}_1(\triangle)| + |\mathcal{L}_2(\triangle)| = |\mathcal{L}_0(\triangle)| + |\mathcal{G}_0(\triangle)| + |\mathcal{H}_0(\triangle)| = a,$$
$$|\mathcal{G}_0(\triangle)| + |\mathcal{G}_1(\triangle)| + |\mathcal{G}_2(\triangle)| = |\mathcal{L}_2(\triangle)| + |\mathcal{G}_2(\triangle)| + |\mathcal{H}_2(\triangle)| = D - f, \tag{10}$$
$$|\mathcal{H}_0(\triangle)| + |\mathcal{H}_1(\triangle)| + |\mathcal{H}_2(\triangle)| = |\mathcal{L}_1(\triangle)| + |\mathcal{G}_1(\triangle)| + |\mathcal{H}_1(\triangle)| = f - a.$$

We will refer this as the intrinsic constraints on the size of above sets.

## A.1    PROOF OF LEMMA 2.1

**Lemma 2.1.** *For any $1 \leq s < t \leq K$ with $t - s = \triangle$, we have*

$$\mathbb{E}_\pi\big[\mathbb{1}\{h_s(\boldsymbol{v}) = h_s(\boldsymbol{w})\}\mathbb{1}\{h_t(\boldsymbol{v}) = h_t(\boldsymbol{w})\}\big] = \frac{|\mathcal{L}_0(\triangle)| + (|\mathcal{G}_0(\triangle)| + |\mathcal{L}_2(\triangle)|)J}{f + |\mathcal{G}_0(\triangle)| + |\mathcal{G}_1(\triangle)|},$$

*where the sets are defined in Definition 2.2 and $h_s$, $h_t$ are the hash values as in Algorithm 2.*

*Proof.* To check whether a hash sample generated by MinHash collides (under some permutation $\pi$), it suffices to look at the permuted location vector $\boldsymbol{x}$. If a collision happens, after permuted by $\pi$, type "$O$" point must appear before the first "$\times$" point. That said, the minimal permutation index of "$O$" elements must be smaller than that of "$\times$" elements. If the hash sample does not collide, then the first "$\times$" must appear before the first "$O$". Note that "$-$" points does not affect the collision.

To compute the variance of the estimator, it suffices to compute $\mathbb{E}[\mathbb{1}_s\mathbb{1}_t]$. Let $\mathcal{L}$, $\mathcal{G}$ and $\mathcal{H}$ be the union of $\mathcal{L}$'s, $\mathcal{G}$'s and $\mathcal{H}$'s, respectively. In the following, we say that an index $i$ belongs to a set if $i$ is the first term of an element in that set. We have

$$|\mathcal{L}| = a, \quad |\mathcal{H}| = f - a, \quad |\mathcal{G}| = D - f.$$

One key observation is that, for a pair $(i,j)$ in above sets, the hash index $\pi_s(i)$ will be the hash index of $\pi_t(j)$. We begin by decomposing the expectation into

$$\mathbb{E}[\mathbb{1}_s\mathbb{1}_t] = P[\text{collision } s, \text{collision } t]$$
$$= \sum_{i_s^* \in \mathcal{L}} P[\text{collision } s \text{ at } i_s^*, \text{collision } t]$$
$$= \sum_{p=0}^{2} \sum_{i_s^* \in \mathcal{L}_p} P[\text{collision } s \text{ at } i_s^*, \text{collision } t]. \tag{11}$$

where $i_s^*$ is the location of the original "$O$" in vector $x$ that collides for $s$-th hash sample. It is different from the exact location of collision in $x(\pi_s)$. Note that the permutation is totally random, so the location of collision is independent of $\mathbb{1}_s$, and uniformly distributed among all type "$O$" pairs.

**1) When $i_s^* \in \mathcal{L}_0$.** In this case, the minimum index of the type "$O$" pair in $x(\pi_s)$, $\pi_s(i_s^*)$, is shifted to another type "$O$" pair in $x(\pi_t)$. Therefore, the indices of pairs with the first element being "$O$" or "$\times$" originally in $x(\pi_s)$ will still be greater than $\pi_t(i_s^*)$. If sample $s$ collides at $i_s^*$, hash sample $t$ will collide when

1. All the points in $\mathcal{G}_1$, after permutation $\pi_s$, is greater than $\pi_s(i_s^*)$. In this case, regardless of the permuted $\mathcal{G}_0$, hash $t$ will always collide.

2. There exist points in $\mathcal{G}_1$ after permutation $\pi_s$ smaller than $\pi_s(i_s^*)$, and also points in $\mathcal{G}_0$ that is smaller than the minimum of permuted $\mathcal{G}_1$.

Consequently, we have for $i_s^* \in \mathcal{L}_0$,

$$P[\text{collision } s \text{ at } i_s^*, \text{collision } t]$$
$$= P[\pi_s(i_s^*) < \pi_s(i), \forall i \in \mathcal{H} \cup \mathcal{L}/i_s^*, \text{ and } \min_{j \in \mathcal{G}_1} \pi_s(j) > \pi_s(i_s^*)]$$
$$\qquad + P[\pi_s(i_s^*) < \pi_s(i), \forall i \in \mathcal{H} \cup \mathcal{L}/i_s^*, \text{ and } \min_{j \in \mathcal{G}_0} \pi_s(j) < \min_{j \in \mathcal{G}_1} \pi_s(j) < \pi_s(i_s^*)]$$
$$= \frac{1}{a} \cdot \frac{a}{f + |\mathcal{G}_1|} + \frac{|\mathcal{G}_0|}{f + |\mathcal{G}_0| + |\mathcal{G}_1|} \cdot \frac{|\mathcal{G}_1|}{f + |\mathcal{G}_1|} \cdot \frac{a}{f} \cdot \frac{1}{a}$$
$$= \frac{1}{f + |\mathcal{G}_1|} + \frac{|\mathcal{G}_0| \cdot |\mathcal{G}_1|}{(f + |\mathcal{G}_0| + |\mathcal{G}_1|)(f + |\mathcal{G}_1|)f}. \tag{12}$$

This probability holds for $\forall i_s^* \in \mathcal{L}_0$.

**2) When $i_s^* \in \mathcal{L}_1$.** Similarly, we consider the condition where $i_s^* \in \mathcal{L}_1$, and both hash samples collide. In this case, $\pi_s(i_s^*)$ would be shifted to a "$\times$" pair in $x(\pi_t)$. That is, the indices of pairs with the first element being "$O$" or "$\times$" originally in $x(\pi_s)$ will all become greater than $\pi_s(i_s^*)$, which now is the location of a "$\times$" pair in $x(\pi_t)$. Thus, to make hash $t$ collide, we will need:

- At least one point from $\mathcal{G}_0$ is smaller than any other points in $\mathcal{H} \cup \mathcal{L} \cup \mathcal{G}_1$ after permutation $\pi_s$.

Therefore, for any $i_s^* \in \mathcal{L}_1$,

$$P[\text{collision } s \text{ at } i_s^*, \text{collision } t]$$
$$= P[\pi_s(i_s^*) < \pi_s(i), \forall i \in \mathcal{H} \cup \mathcal{L}/i_s^*, \text{ and } \min_{j \in \mathcal{G}_0} \pi_s(j) < \min\{\pi_s(i_s^*), \min_{j \in \mathcal{G}_1} \pi_s(j)\}]$$
$$= \frac{|\mathcal{G}_0|}{f + |\mathcal{G}_0| + |\mathcal{G}_1|} \cdot \frac{a}{f} \cdot \frac{1}{a}$$
$$= \frac{|\mathcal{G}_0|}{(f + |\mathcal{G}_0| + |\mathcal{G}_1|)f}, \tag{13}$$

which is true for $\forall i_s^* \in \mathcal{L}_1$.

**3) When $i_s^* \in \mathcal{L}_2$.**

In this scenario, $\pi_s(i_s^*)$ would be shifted to a "$-$" pair in $x(\pi_t)$. Therefore, if hash $s$ collides, hash $t$ will also collide when:

- After applying $\pi_s$, the minimum of $\mathcal{L}_0 \cup \mathcal{H}_0 \cup \mathcal{G}_0$ is smaller than the minimum of $\mathcal{L}_1 \cup \mathcal{H}_1 \cup \mathcal{G}_1$.

Thus, we obtain that for any $i_s^* \in \mathcal{L}_2$,

$$P[\text{collision } s \text{ at } i_s^*, \text{collision } t]$$
$$= P[\pi_s(i_s^*) < \pi_s(i), \forall i \in \mathcal{H} \cup \mathcal{L}/i_s^*, \text{ and } \min_{j \in \mathcal{L}_0 \cup \mathcal{G}_0 \cup \mathcal{H}_0} \pi_s(j) < \min_{j \in \mathcal{L}_1 \cup \mathcal{G}_1 \cup \mathcal{H}_1} \pi_s(j)]$$
$$\triangleq P[\Omega].$$

Let $\mathbb{1}_{s,i_s^*}$ denote the event $\{\pi_s(i_s^*) < \pi_s(i), \forall i \in \mathcal{H} \cup \mathcal{L}/i_s^*\}$. Then $\Omega$ can be separated into the following several cases:

1. $\Omega_1$: $\mathbb{1}_{s,i_s^*}$, $\min_{j \in \mathcal{L}_0 \cup \mathcal{H}_0} \pi_s(j) < \min_{j \in \mathcal{L}_1 \cup \mathcal{H}_1} \pi_s(j)$, and $\min_{j \in \mathcal{L}_0 \cup \mathcal{H}_0} \pi_s(j) < \min_{j \in \mathcal{G}_1} \pi_s(j)$.

2. $\Omega_2$: $\mathbb{1}_{s,i_s^*}$, $\min_{j \in \mathcal{L}_0 \cup \mathcal{H}_0} \pi_s(j) < \min_{j \in \mathcal{L}_1 \cup \mathcal{H}_1} \pi_s(j)$, and $\min_{j \in \mathcal{L}_0 \cup \mathcal{H}_0} \pi_s(j) > \min_{j \in \mathcal{G}_1} \pi_s(j) > \min_{j \in \mathcal{G}_0} \pi_s(j) > \pi_s(i_s^*)$.

3. $\Omega_3$: $\mathbb{1}_{s,i_s^*}$, $\min_{j \in \mathcal{L}_0 \cup \mathcal{H}_0} \pi_s(j) < \min_{j \in \mathcal{L}_1 \cup \mathcal{H}_1} \pi_s(j)$, and $\min_{j \in \mathcal{L}_0 \cup \mathcal{H}_0} \pi_s(j) > \min_{j \in \mathcal{G}_1} \pi_s(j) > \pi_s(i_s^*) > \min_{j \in \mathcal{G}_0} \pi_s(j)$.

4. $\Omega_4$: $\mathbb{1}_{s,i_s^*}$, $\min_{j \in \mathcal{L}_0 \cup \mathcal{H}_0} \pi_s(j) < \min_{j \in \mathcal{L}_1 \cup \mathcal{H}_1} \pi_s(j)$, and $\min_{j \in \mathcal{L}_0 \cup \mathcal{H}_0} \pi_s(j) > \pi_s(i_s^*) > \min_{j \in \mathcal{G}_1} \pi_s(j) > \min_{j \in \mathcal{G}_0} \pi_s(j)$.

5. $\Omega_5$: $\mathbb{1}_{s,i_s^*}$, $\min_{j \in \mathcal{L}_0 \cup \mathcal{H}_0} \pi_s(j) > \min_{j \in \mathcal{L}_1 \cup \mathcal{H}_1} \pi_s(j)$, and $\pi_s(i_s^*) < \min_{j \in \mathcal{G}_0} \pi_s(j) < \min_{j \in \mathcal{L}_1 \cup \mathcal{H}_1 \cup \mathcal{G}_1} \pi_s(j)$.

6. $\Omega_6$: $\mathbb{1}_{s,i_s^*}$, $\min_{j \in \mathcal{L}_0 \cup \mathcal{H}_0} \pi_s(j) > \min_{j \in \mathcal{L}_1 \cup \mathcal{H}_1} \pi_s(j)$, and $\min_{j \in \mathcal{G}_0} \pi_s(j) < \pi_s(i_s^*) < \min_{j \in \mathcal{L}_1 \cup \mathcal{H}_1 \cup \mathcal{G}_1} \pi_s(j)$.

We can compute the probability of each event as

$$
\begin{aligned}
P[\Omega_1] &= \frac{1}{a} \cdot \frac{a}{f + |\mathcal{G}_1|} \cdot \frac{|\mathcal{L}_0| + |\mathcal{H}_0|}{|\mathcal{L}_0| + |\mathcal{H}_0| + |\mathcal{L}_1| + |\mathcal{H}_1| + |\mathcal{G}_1|}, \\
&= \frac{a - |\mathcal{G}_0|}{(f - |\mathcal{G}_0|)(f + |\mathcal{G}_1|)}, \\
P[\Omega_2] &= \frac{1}{a} \cdot \frac{a}{f + |\mathcal{G}_0| + |\mathcal{G}_1|} \cdot \frac{|\mathcal{G}_0|}{|\mathcal{G}_0| + |\mathcal{G}_1| + |\mathcal{L}_0| + |\mathcal{H}_0| + |\mathcal{L}_1| + |\mathcal{H}_1|} \\
&\quad \cdot \frac{|\mathcal{G}_1|}{|\mathcal{L}_0| + |\mathcal{H}_0| + |\mathcal{L}_1| + |\mathcal{H}_1| + |\mathcal{G}_1|} \cdot \frac{|\mathcal{L}_0| + |\mathcal{H}_0|}{|\mathcal{L}_0| + |\mathcal{H}_0| + |\mathcal{L}_1| + |\mathcal{H}_1|} \\
&= \frac{1}{f + |\mathcal{G}_0| + |\mathcal{G}_1|} \cdot \frac{|\mathcal{G}_0|}{f} \cdot \frac{|\mathcal{G}_1|}{f - |\mathcal{G}_0|} \cdot \frac{a - |\mathcal{G}_0|}{f - |\mathcal{G}_0| - |\mathcal{G}_1|} \\
&= \frac{|\mathcal{G}_0| \cdot |\mathcal{G}_1| \cdot (a - |\mathcal{G}_0|)}{(f + |\mathcal{G}_0| + |\mathcal{G}_1|)(f - |\mathcal{G}_0|)(f - |\mathcal{G}_0| - |\mathcal{G}_1|)f}, \\
P[\Omega_3] &= \frac{|\mathcal{G}_0|}{f + |\mathcal{G}_0| + |\mathcal{G}_1|} \cdot \frac{1}{f + |\mathcal{G}_1|} \cdot \frac{|\mathcal{G}_1|}{|\mathcal{L}_0| + |\mathcal{H}_0| + |\mathcal{L}_1| + |\mathcal{H}_1| + |\mathcal{G}_1|} \cdot \frac{|\mathcal{L}_0| + |\mathcal{H}_0|}{|\mathcal{L}_0| + |\mathcal{H}_0| + |\mathcal{L}_1| + |\mathcal{H}_1|} \\
&= \frac{|\mathcal{G}_0| \cdot |\mathcal{G}_1| \cdot (a - |\mathcal{G}_0|)}{(f + |\mathcal{G}_0| + |\mathcal{G}_1|)(f + |\mathcal{G}_1|)(f - |\mathcal{G}_0|)(f - |\mathcal{G}_0| - |\mathcal{G}_1|)}, \\
P[\Omega_4] &= \frac{|\mathcal{G}_0|}{f + |\mathcal{G}_0| + |\mathcal{G}_1|} \cdot \frac{|\mathcal{G}_1|}{f + |\mathcal{G}_1|} \cdot \frac{1}{f} \cdot \frac{|\mathcal{L}_0| + |\mathcal{H}_0|}{|\mathcal{L}_0| + |\mathcal{H}_0| + |\mathcal{L}_1| + |\mathcal{H}_1|} \\
&= \frac{|\mathcal{G}_0| \cdot |\mathcal{G}_1| \cdot (a - |\mathcal{G}_0|)}{(f + |\mathcal{G}_0| + |\mathcal{G}_1|)(f + |\mathcal{G}_1|)(f - |\mathcal{G}_0| - |\mathcal{G}_1|)f}, \\
P[\Omega_5] &= \frac{1}{f + |\mathcal{G}_0| + |\mathcal{G}_1|} \cdot \frac{|\mathcal{G}_0|}{|\mathcal{G}_0| + |\mathcal{G}_1| + |\mathcal{L}_0| + |\mathcal{H}_0| + |\mathcal{L}_1| + |\mathcal{H}_1|} \cdot \frac{|\mathcal{L}_1| + |\mathcal{H}_1|}{|\mathcal{L}_0| + |\mathcal{H}_0| + |\mathcal{L}_1| + |\mathcal{H}_1|} \\
&= \frac{|\mathcal{G}_0| \cdot (f - a - |\mathcal{G}_1|)}{(f + |\mathcal{G}_0| + |\mathcal{G}_1|)(f - |\mathcal{G}_0| - |\mathcal{G}_1|)f}, \\
P[\Omega_6] &= \frac{|\mathcal{G}_0|}{f + |\mathcal{G}_0| + |\mathcal{G}_1|} \cdot \frac{1}{f} \cdot \frac{|\mathcal{L}_1| + |\mathcal{H}_1|}{|\mathcal{L}_0| + |\mathcal{H}_0| + |\mathcal{L}_1| + |\mathcal{H}_1|} \\
&= \frac{|\mathcal{G}_0| \cdot (f - a - |\mathcal{G}_1|)}{(f + |\mathcal{G}_0| + |\mathcal{G}_1|)(f - |\mathcal{G}_0| - |\mathcal{G}_1|)f}.
\end{aligned}
$$

Note that

$$P[\Omega_2] + P[\Omega_3] + P[\Omega_4]$$

$$= \frac{|\mathcal{G}_0| \cdot |\mathcal{G}_1| \cdot (a - |\mathcal{G}_0|)}{(f + |\mathcal{G}_0| + |\mathcal{G}_1|)(f - |\mathcal{G}_0| - |\mathcal{G}_1|)} \left[ \frac{1}{(f - |\mathcal{G}_0|)f} + \frac{1}{(f - |\mathcal{G}_0|)(f + |\mathcal{G}_1|)} + \frac{1}{f(f + |\mathcal{G}_1|)} \right]$$

$$= \frac{|\mathcal{G}_0| \cdot |\mathcal{G}_1| \cdot (a - |\mathcal{G}_0|)(3f - |\mathcal{G}_0| + |\mathcal{G}_1|)}{(f + |\mathcal{G}_0| + |\mathcal{G}_1|)(f - |\mathcal{G}_0| - |\mathcal{G}_1|)f(f - |\mathcal{G}_0|)(f + |\mathcal{G}_1|)}.$$

Summing up all the terms together, we obtain $P[\Omega]$ as

$$\sum_{n=1}^{6} P[\Omega_n] = \frac{f(f + |\mathcal{G}_0| + |\mathcal{G}_1|)(f - |\mathcal{G}_0| - |\mathcal{G}_1|)(a - |\mathcal{G}_0|) + |\mathcal{G}_0||\mathcal{G}_1|(a - |\mathcal{G}_0|)(3f - |\mathcal{G}_0| + |\mathcal{G}_1|)}{(f + |\mathcal{G}_0| + |\mathcal{G}_1|)(f - |\mathcal{G}_0| - |\mathcal{G}_1|)(f - |\mathcal{G}_0|)(f + |\mathcal{G}_1|)f}$$

$$+ \frac{2|\mathcal{G}_0|(f - a - |\mathcal{G}_1|)(f - |\mathcal{G}_0|)(f + |\mathcal{G}_1|)}{(f + |\mathcal{G}_0| + |\mathcal{G}_1|)(f - |\mathcal{G}_0| - |\mathcal{G}_1|)(f - |\mathcal{G}_0|)(f + |\mathcal{G}_1|)f}$$

$$= \frac{(a - |\mathcal{G}_0|)(f + |\mathcal{G}_0| - |\mathcal{G}_1|)(f - |\mathcal{G}_0|)(f + |\mathcal{G}_1|) + 2|\mathcal{G}_0|(f - a - |\mathcal{G}_1|)(f - |\mathcal{G}_0|)(f + |\mathcal{G}_1|)}{(f + |\mathcal{G}_0| + |\mathcal{G}_1|)(f - |\mathcal{G}_0| - |\mathcal{G}_1|)(f - |\mathcal{G}_0|)(f + |\mathcal{G}_1|)f}$$

$$= \frac{(a + |\mathcal{G}_0|)(f - |\mathcal{G}_0| - |\mathcal{G}_1|)(f - |\mathcal{G}_0|)(f + |\mathcal{G}_1|)}{(f + |\mathcal{G}_0| + |\mathcal{G}_1|)(f - |\mathcal{G}_0| - |\mathcal{G}_1|)(f - |\mathcal{G}_0|)(f + |\mathcal{G}_1|)f}$$

$$= \frac{a + |\mathcal{G}_0|}{(f + |\mathcal{G}_0| + |\mathcal{G}_1|)f}, \tag{14}$$

which holds for $\forall i_s^* \in \mathcal{L}_2$. Now combining (12), (13), (14) with (11), we obtain

$$\mathbb{E}[\mathbb{1}_s \mathbb{1}_t] = \frac{|\mathcal{L}_0|}{f + |\mathcal{G}_1|} + \frac{|\mathcal{G}_0||\mathcal{G}_1||\mathcal{L}_0|}{(f + |\mathcal{G}_0| + |\mathcal{G}_1|)(f + |\mathcal{G}_1|)f} + \frac{|\mathcal{G}_0||\mathcal{L}_1|}{(f + |\mathcal{G}_0| + |\mathcal{G}_1|)f} + \frac{(a + |\mathcal{G}_0|)|\mathcal{L}_2|}{(f + |\mathcal{G}_0| + |\mathcal{G}_1|)f}. \tag{15}$$

Here, recall that the sets are associated with all $1 \le s < t \le K$ such that $\triangle = t - s$. Using the intrinsic constraints (10), after some calculation we can simplify (15) as

$$\mathbb{E}_\pi[\mathbb{1}_s \mathbb{1}_t] = \frac{|\mathcal{L}_0|}{f + |\mathcal{G}_0| + |\mathcal{G}_1|} + \frac{a(|\mathcal{G}_0| + |\mathcal{L}_2|)}{(f + |\mathcal{G}_0| + |\mathcal{G}_1|)f},$$

which completes the proof. $\qquad \square$

## A.2 PROOF OF THEOREM 2.2

**Theorem 2.2.** *Under the same setting as in Lemma 2.1, the variance of $\hat{J}_{0,\pi}$ is*

$$Var[\hat{J}_{0,\pi}] = \frac{J}{K} + \frac{2 \sum_{s=2}^{K}(s-1)\Theta_{K-s+1}}{K^2} - J^2,$$

*where $\Theta_\triangle \triangleq E_\pi\big[\mathbb{1}\{h_s(\boldsymbol{v}) = h_s(\boldsymbol{w})\}\mathbb{1}\{h_t(\boldsymbol{v}) = h_t(\boldsymbol{w})\}\big]$ as in Lemma 2.1 with any $t - s = \triangle$.*

*Proof.* By the expansion of variance formula, since $\mathbb{E}[\mathbb{1}_s^2] = \mathbb{E}[\mathbb{1}_s] = J$, we have

$$Var[\hat{J}_{0,\pi}] = \frac{J}{K} + \frac{\sum_{s=1}^{K}\sum_{t \ne s}^{K}\mathbb{E}[\mathbb{1}_s \mathbb{1}_t]}{K^2} - J^2. \tag{16}$$

Note here that for $\forall t > s$, the $t$-th hash sample uses $\pi_t$ as the permutation, which is shifted rightwards by $\triangle = t - s$ from $\pi_s$. Thus, we have $\mathbb{E}[\mathbb{1}_s \mathbb{1}_t] = \mathbb{E}[\mathbb{1}_{s-i} \mathbb{1}_{t-i}]$ for $\forall 0 < i < s \wedge t$, which implies $\mathbb{E}[\mathbb{1}_s \mathbb{1}_t] = \mathbb{E}[\mathbb{1}_1 \mathbb{1}_{t-s+1}]$, $\forall s < t$. Since by assumption $K \le D$, we have

$$\sum_{s}^{K}\sum_{t \ne s}^{K}\mathbb{E}[\mathbb{1}_s \mathbb{1}_t] = 2\mathbb{E}\big[(\mathbb{1}_1\mathbb{1}_2 + \mathbb{1}_1\mathbb{1}_3 + ... + \mathbb{1}_1\mathbb{1}_K) + (\mathbb{1}_2\mathbb{1}_3 + ... + \mathbb{1}_2\mathbb{1}_K) + ... + \mathbb{1}_{K-1}\mathbb{1}_K\big]$$

$$= 2\mathbb{E}\big[(\mathbb{1}_1\mathbb{1}_2 + \mathbb{1}_1\mathbb{1}_3 + ... + \mathbb{1}_1\mathbb{1}_K) + (\mathbb{1}_1\mathbb{1}_2 + ... + \mathbb{1}_1\mathbb{1}_{K-1}) + ... + \mathbb{1}_1\mathbb{1}_2\big]$$

$$= 2 \sum_{s=2}^{K} (s-1) \mathbb{E}[\mathbb{1}_1 \mathbb{1}_{K-s+2}]$$

$$\triangleq 2 \sum_{s=2}^{K} (s-1) \Theta_{K-s+1}. \tag{17}$$

Finally, integrating (16), (17) and Lemma 2.1 completes the proof. □

### A.3 PROOF OF THEOREM 3.2

**Theorem 3.2.** *Let $a, f$ be defined as in (5). When $0 < a < f \le D$ ($J \notin \{0,1\}$), we have*

$$Var[\hat{J}_{\sigma,\pi}] = \frac{J}{K} + \frac{(K-1)\tilde{\mathcal{E}}}{K} - J^2, \tag{18}$$

*where with $l = \max(0, D - 2f + a)$, and*

$$\tilde{\mathcal{E}} = \sum_{\{l_0, l_2, g_0, g_1\}} \left\{ \left( \frac{l_0}{f + g_0 + g_1} + \frac{a(g_0 + l_2)}{(f + g_0 + g_1)f} \right) \right.$$
$$\left. \times \sum_{s=l}^{D-f-1} \frac{\binom{D-f}{s}}{\binom{D-a-1}{D-f-1}} \frac{\binom{f-a-1}{D-f-s-1}\binom{s}{n_1}\binom{D-f-s}{n_2}\binom{D-f-s}{n_3}\binom{f-a-(D-f-s)}{n_4}\binom{a-1}{a-l_1-l_2}}{\binom{D-1}{a}} \right\}. \tag{19}$$

*The feasible set $\{l_0, l_2, g_0, g_1\}$ satisfies the intrinsic constraints (6), and*

$$n_1 = g_0 - (D - f - s - g_1), \qquad n_2 = D - f - s - g_1,$$
$$n_3 = l_2 - g_0 + (D - f - s - g_1), \quad n_4 = l_1 - (D - f - s - g_1).$$

*Also, when $a = 0$ or $f = a$ ($J = 0$ or $J = 1$), we have $Var[\hat{J}_{\sigma,\pi}] = 0$.*

*Proof.* Similar to the proof of Theorem 2.2, we denote $\Theta_\triangle = \mathbb{E}_{\sigma,\pi}[\mathbb{1}_s \mathbb{1}_t]$ with $|t - s| = \triangle$. Note that now the expectation is taken w.r.t. both two independent permutations $\sigma$ and $\pi$. Since $\sigma$ is random, we know that $\Theta_1 = \Theta_2 = \cdots = \Theta_{K-1}$. Then by the variance formula, we have

$$Var[\hat{J}_{\sigma,\pi}] = \frac{J^2}{K} - \frac{(K-1)\Theta_1}{K} - J^2 \tag{20}$$

Hence, it suffices to consider $\Theta_1$. In this proof, we will set $\triangle = 1$ and drop the notation $\triangle$ for conciseness, and denote $\tilde{\mathcal{E}} = \Theta_1$ from now on. First, we note that Lemma 2.1 gives the desired quantity conditional on $\sigma$. By the law of total probability, we have

$$\tilde{\mathcal{E}} = \mathbb{E}_\sigma \left[ \frac{|\mathcal{L}_0|}{f + |\mathcal{G}_0| + |\mathcal{G}_1|} + \frac{a(|\mathcal{G}_0| + |\mathcal{L}_2|)}{(f + |\mathcal{G}_0| + |\mathcal{G}_1|)f} \right], \tag{21}$$

where the sizes of sets are random depending on the initial permutation $\sigma$ (i.e. counted after permuting by $\sigma$). As a result, the problem turns into deriving the distribution of $|\mathcal{L}_0|, |\mathcal{L}_1|, |\mathcal{L}_2|, |\mathcal{G}_0|$ and $|\mathcal{G}_1|$ under random permutation $\sigma$, and then taking expectation of (21) with respect to this additional randomness.

When $a = 0$, we know that $|\mathcal{L}_0| = |\mathcal{L}_2| = |\mathcal{G}_0| = 0$, hence the expectation $\tilde{\mathcal{E}}$ is trivially 0. Thus, the $Var[\hat{J}_{\sigma,\pi}] = 0$. When $f = a$, $|\mathcal{G}_1| = 0$, and the constraint on the sets becomes

$$|\mathcal{L}_0| + |\mathcal{G}_0| = |\mathcal{L}_0| + |\mathcal{L}_2| = f,$$
$$|\mathcal{L}_2| + |\mathcal{G}_2| = |\mathcal{G}_0| + |\mathcal{G}_2| = D - f.$$

Then (21) becomes

$$\tilde{\mathcal{E}} = \mathbb{E}_\sigma \left[ \frac{|\mathcal{L}_0|}{f + |\mathcal{G}_0|} + \frac{|\mathcal{G}_0| + |\mathcal{L}_2|}{f + |\mathcal{G}_0|} \right]$$

$$= \mathbb{E}_\sigma \left[ \frac{|\mathcal{L}_0| + |\mathcal{G}_0| + |\mathcal{L}_2|}{f + |\mathcal{G}_0|} \right] \equiv 1.$$

Therefore, when $f = a$, we also have $Var[\hat{J}_{\sigma,\pi}] = 0$.

Next, we will consider the general case where $0 < a < f \leq D$. This can be considered as a combinatorial problem where we randomly arrange $a$ type "$O$", $(f - a)$ type "$\times$" and $(D - f)$ type "$-$" points in a circle. We are interested in the distribution of the number of $\{O, O\}$, $\{O, \times\}$, $\{O, -\}$, $\{-, O\}$ and $\{-, \times\}$ pairs of consecutive points in clockwise direction. We consider this procedure in two steps, where we first place "$\times$" and "$-$" points, and then place "$O$" points.

**Step 1. Randomly place "$\times$" and "$-$" points on the circle.**

In this step, four types of pairs may appear: $\{-, -\}$, $\{-, \times\}$, $\{\times, \times\}$ and $\{\times, -\}$. Denote $\mathcal{C}_1$, $\mathcal{C}_2$, $\mathcal{C}_3$ and $\mathcal{C}_4$ as the collections of above pairs. Since

$$|\mathcal{C}_1| + |\mathcal{C}_4| = |\mathcal{C}_1| + |\mathcal{C}_2| = D - f,$$
$$|\mathcal{C}_2| + |\mathcal{C}_3| = |\mathcal{C}_2| + |\mathcal{C}_4| = f - a,$$

knowing the size of one set gives information on the size of all the sets. Thus, we can characterize the joint distribution by analyzing the distribution of $|\mathcal{C}_1|$. First, placing $(D - f)$ "$-$" points on a circle leads to $(D - f)$ number of $\{-, -\}$ pairs. This $(D - f)$ elements can be regarded as the borders that split the circle into $(D - f)$ bins. Now, we randomly throw $(f - a)$ number of "$\times$" points into these bins. If at least one "$\times$" falls into one bin, then the number of $\{-, -\}$ pairs ($|\mathcal{C}_1|$) would reduce by 1, while $|\mathcal{C}_2|$ and $|\mathcal{C}_4|$ would increase by 1. If $z$ "$\times$" points fall into one bin, then the number of $\{\times, \times\}$ ($|\mathcal{C}_3|$) would increase by $(z - 1)$. Notice that since $s \leq D - f$ and $D - f - s \leq f - a$, we have $\max(0, D - 2f + a) \leq s \leq D - f$. Consequently, for $s$ in this range, we have

$$P\Big\{|\mathcal{C}_1| = s\Big\} = P\Big\{|\mathcal{C}_1| = s, |\mathcal{C}_3| = f - a - (D - f - s)\Big\}$$
$$= \frac{\binom{D-f}{D-f-s}\binom{f-a-1}{D-f-s-1}}{\binom{D-a-1}{D-f-1}}$$
$$= \frac{\binom{D-f}{s}\binom{f-a-1}{D-f-s-1}}{\binom{D-a-1}{D-f-1}}. \tag{22}$$

The second line is due to the stars and bars problem that the number of ways to place $n$ unlabeled balls in $m$ distinct bins such that each bin has at least one ball is $\binom{n-1}{m-1}$. For $|\mathcal{C}_1| = s$, we need $n = f - a$ (number of "$\times$") and $m = |\mathcal{C}_2| = D - f - s$. Moreover, the number of ways to place $n$ balls in $m$ distinct bins is $\binom{n+m-1}{m-1}$. When counting the total number of possibilities, we have $n = f - a$ and $m = D - f$. This gives the denominator. We notice that (22) is actually a hyper-geometric distribution.

**Step 2. Randomly place "$O$" points on the circle.**

We have the probability mass function

$$P[\Psi] \triangleq P\Big\{|\mathcal{L}_1| = l_1, |\mathcal{L}_2| = l_2, |\mathcal{G}_0| = g_0, |\mathcal{G}_1| = g_1\Big\}$$
$$= \sum_{s=D-2f+a}^{D-f-1} P\Big\{|\mathcal{L}_1| = l_1, |\mathcal{L}_2| = l_2, |\mathcal{G}_0| = g_0, |\mathcal{G}_1| = g_1 \Big| |\mathcal{C}_1| = s\Big\} P\Big\{|\mathcal{C}_1| = s\Big\}. \tag{23}$$

Now it remains to compute the distribution conditional on $|\mathcal{C}_1|$. Here we drop $|\mathcal{L}_0|$ because it is intrinsically determined by $|\mathcal{L}_1|$ and $|\mathcal{L}_2|$. Again, given a placement of all "$\times$" and "$-$" points, each consecutive pair can be regarded as a distinct bin. Therefore, now the problem is to randomly throw $a$ type "$O$" points into that $(D - a)$ bins, given that we have placed type "$\times$" and "$-$" points on the circle with $|\mathcal{C}_1| = s$ (and thus $|\mathcal{C}_2| = |\mathcal{C}_3| = D - f - s$ and $|\mathcal{C}_4| = f - a - (D - f - s)$ are also determined correspondingly). In the following, we count the number of "$O$" points that fall in $\mathcal{C}_i$, $i = 1, 2, 3, 4$, to make the event $\Psi$ happen. Note that

- When at least one "$O$" point falls into $\mathcal{C}_1$ (between $\{-, -\}$), $|\mathcal{L}_2|$ and $|\mathcal{G}_0|$ increase by 1.

- When at least one "$O$" point falls into $\mathcal{C}_2$ (between $\{-, \times\}$), $|\mathcal{L}_1|$ and $|\mathcal{G}_0|$ increase by 1, while $|\mathcal{G}_1|$ decreases by 1.

- When at least one "$O$" point falls into $\mathcal{C}_3$ (between $\{\times, -\}$), $|\mathcal{L}_2|$ increases by 1.

- When at least one "$O$" point falls into $\mathcal{C}_4$ (between $\{\times, \times\}$), $|\mathcal{L}_1|$ increases by 1.

We denote the number of bins in $\mathcal{C}_i$, $i = 1, 2, 3, 4$ that contain at least one "$O$" point as $n_1, n_2, n_3, n_4$, respectively. As a result of above reasoning, in the event $\Psi$, we have

$$
\begin{cases}
n_1 + n_3 = l_2, \\
n_2 + n_4 = l_1, \\
n_1 + n_2 = g_0, \\
D - f - s - n_2 = g_1.
\end{cases}
$$

Solving the equations gives

$$
\begin{cases}
n_1 = g_0 - (D - f - s - g_1), \\
n_2 = D - f - s - g_1, \\
n_3 = l_2 - g_0 + (D - f - s - g_1), \\
n_4 = l_1 - (D - f - s - g_1).
\end{cases}
$$

Note that $\sum_{i=1}^4 n_i = l_1 + l_2$. Therefore, event $\Psi$ is equivalent to randomly pick $n_1, n_2, n_3$ and $n_4$ bins in $\mathcal{C}_1,...,\mathcal{C}_4$, and then distribute $a$ type "$O$" points in these $(l_1 + l_2)$ bins such that each bin contains at least one "$O$". Hence, we obtain

$$
\begin{aligned}
P\Big\{ |\mathcal{L}_1| = l_1, |\mathcal{L}_2| = l_2, |\mathcal{G}_0| = g_0, |\mathcal{G}_1| = g_1 \Big| |\mathcal{C}_1| = s \Big\} &= \frac{\binom{s}{n_1}\binom{D-f-s}{n_2}\binom{D-f-s}{n_3}\binom{f-a-(D-f-s)}{n_4}\binom{a-1}{l_1+l_2-1}}{\binom{D-1}{D-a-1}} \\
&= \frac{\binom{s}{n_1}\binom{D-f-s}{n_2}\binom{D-f-s}{n_3}\binom{f-a-(D-f-s)}{n_4}\binom{a-1}{a-l_1-l_2}}{\binom{D-1}{a}},
\end{aligned}
\tag{24}
$$

which is also multi-variate hyper-geometric distributed. Now combining (22), (23) and (24), we obtain the joint distribution of $|\mathcal{L}_0|, |\mathcal{L}_1|, |\mathcal{L}_2|, |\mathcal{G}_0|$ and $|\mathcal{G}_1|$ as

$$
\begin{aligned}
&P\Big\{ |\mathcal{L}_1| = l_1, |\mathcal{L}_2| = l_2, |\mathcal{G}_0| = g_0, |\mathcal{G}_1| = g_1 \Big\} \\
&= \sum_{s=\max(0, D-2f+a)}^{D-f-1} \frac{\binom{s}{n_1}\binom{D-f-s}{n_2}\binom{D-f-s}{n_3}\binom{f-a-(D-f-s)}{n_4}\binom{a-1}{a-l_1-l_2}}{\binom{D-1}{a}} \cdot \frac{\binom{D-f}{s}\binom{f-a-1}{D-f-s-1}}{\binom{D-a-1}{D-f-1}}.
\end{aligned}
\tag{25}
$$

Now let $\Xi$ be the feasible set of $(l_0, l_1, g_0, g_1, g_2)$ that satisfies the intrinsic constraints (10). The desired expectation w.r.t. both $\pi$ and $\sigma$ can thus be written as

$$
\begin{aligned}
\tilde{\mathcal{E}} = \sum_{\Xi} &\left( \frac{l_0}{f + g_0 + g_1} + \frac{a(g_0 + l_2)}{(f + g_0 + g_1)f} \right) \cdot \\
&\left( \sum_{s=\max(0, D-2f+a)}^{D-f-1} \frac{\binom{s}{n_1}\binom{D-f-s}{n_2}\binom{D-f-s}{n_3}\binom{f-a-(D-f-s)}{n_4}\binom{a-1}{a-l_1-l_2}}{\binom{D-1}{a}} \cdot \frac{\binom{D-f}{s}\binom{f-a-1}{D-f-s-1}}{\binom{D-a-1}{D-f-1}} \right).
\end{aligned}
$$

The desired result can then follows by (20). □

## A.4 PROOF OF PROPOSITION 3.3

**Proposition 3.3** (Symmetry). $Var[\hat{J}_{\sigma, \pi}]$ *is the same for the* $(D, f, a)$-*data pair and the* $(D, f, f - a)$-*data pair,* $\forall 0 \le a \le f \le D$.

*Proof.* For fixed $a, f, D$, let $\tilde{\mathcal{E}}_1$ be the expectation defined in Theorem 3.2 for $(\boldsymbol{v}_1, \boldsymbol{w}_1)$, and $\tilde{\mathcal{E}}_2$ be that for $(\boldsymbol{v}_2, \boldsymbol{w}_2)$. From Theorem 3.2 we know that

$$\tilde{\mathcal{E}}_1 = \mathbb{E}_{(l_0, l_2, g_0, g_1)}\Big[\frac{l_0}{f + g_0 + g_1} + \frac{a(g_0 + l_2)}{(f + g_0 + g_1)f}\Big],$$

where $(l_0, l_2, g_0, g_1)$ follows the distribution of $(|\mathcal{L}_0|, |\mathcal{L}_2|, |\mathcal{G}_0|, |\mathcal{G}_1|)$ associated with the location vector $\boldsymbol{x}_1$ of $(\boldsymbol{v}_1, \boldsymbol{v}_2)$. For data pair $(\boldsymbol{v}_2, \boldsymbol{w}_2)$, we can consider its location vector $\boldsymbol{x}_2$ as swapping the "$O$" and "$\times$" entries of $\boldsymbol{x}_1$. Now we denote the size of the corresponding sets (Definition 2.2) of $\boldsymbol{x}_2$ as $l_i's, g_i's, h_i's$, for $i = 0, 1, 2$. Since $\sigma$ is applied before hashing, by symmetry there is a one-to-one correspondence between the two location vectors. More specifically, $l_0'$ corresponds to $h_1$, $g_0'$ corresponds to $g_1$, $g_1'$ corresponds to $g_0$, and $l_2'$ corresponds to $h_2$. Therefore, in probability we can write

$$\tilde{\mathcal{E}}_2 = \mathbb{E}_{(l_0', l_2', g_0', g_1')}\Big[\frac{l_0'}{f + g_0' + g_1'} + \frac{a(g_0' + l_2')}{(f + g_0' + g_1')f}\Big]$$

$$= \mathbb{E}_{(h_1, h_2, g_0, g_1)}\Big[\frac{h_1}{f + g_0 + g_1} + \frac{(f - a)(g_1 + h_2)}{(f + g_0 + g_1)f}\Big].$$

Consequently, we have

$$\tilde{\mathcal{E}}_1 - \tilde{\mathcal{E}}_2 = \mathbb{E}_{(l_0, l_2, h_1, h_2, g_0, g_1)}\Big[\frac{l_0 - h_1}{f + g_0 + g_1} + \frac{a(g_0 + l_2) - (f - a)(g_1 + h_2)}{(f + g_0 + g_1)f}\Big].$$

In the sequel, the subscript of expectation is suppressed for conciseness. Exploiting the constraints (10), we deduce that $h_1 = (f - a) - l_1 - g_1$, $h_2 = l_0 + g_0 + l_1 + g_1 - a$ and $l_0 + l_1 = a - l_2$. Using these facts we obtain

$$\tilde{\mathcal{E}}_1 - \tilde{\mathcal{E}}_2 = \mathbb{E}\Big[\frac{(l_0 - (f - a) + l_1 + g_1)f + a(g_0 + l_2) - (f - a)(l_0 + g_0 + l_1 + 2g_1 - a)}{(f + g_0 + g_1)f}\Big]$$

$$= \mathbb{E}\Big[\frac{(2a - f + g_1 - l_2)f + a(g_0 + l_2) - (f - a)(2g_1 + g_0 - l_2)}{(f + g_0 + g_1)f}\Big]$$

$$= \mathbb{E}\Big[\frac{2(f + g_0 + g_1)a - (f + g_0 + g_1)f}{(f + g_0 + g_1)f}\Big]$$

$$= 2J - 1.$$

Comparing the variances of $\hat{J}_{\sigma, \pi}(\boldsymbol{v}_1, \boldsymbol{w}_1)$ and $\hat{J}_{\sigma, \pi}(\boldsymbol{v}_2, \boldsymbol{w}_2)$, we derive

$$Var[\hat{J}_{\sigma, \pi}(\boldsymbol{v}_1, \boldsymbol{w}_1)] - Var[\hat{J}_{\sigma, \pi}(\boldsymbol{v}_2, \boldsymbol{w}_2)]$$

$$= (\frac{J}{K} + \frac{(K - 1)\tilde{\mathcal{E}}_1}{K} - J^2) - (\frac{1 - J}{K} + \frac{(K - 1)\tilde{\mathcal{E}}_2}{K} - (1 - J)^2)$$

$$= -\frac{K - 1}{K}(2J - 1) + \frac{K - 1}{K}(\tilde{\mathcal{E}}_1 - \tilde{\mathcal{E}}_2) = 0.$$

This completes the proof. $\square$

## A.5    PROOF OF LEMMA 3.4

**Lemma 3.4** (Increasing Increment). *Assume $a > 0$ and $f > a$ are arbitrary and fixed. Denote $\tilde{\mathcal{E}}_D$ as in (19) in Theorem 3.2, with $D$ treated as a parameter. Then we have $\tilde{\mathcal{E}}_{D+1} > \tilde{\mathcal{E}}_D$ for $\forall D \geq f$.*

*Proof.* Let the probability mass function (25) with $a$, $f$ and dimension $D$ be $P_{a, f, D}(l_0, l_2, g_0, g_1)$. Conditional on $l_0, l_2, g_0, g_1$ with $D$ elements, the possible values $l_0', l_2', g_0', g_1'$ when adding a "$-$" are

- $g_0' = g_0 + 1, l_0' = l_0, l_2' = l_2, g_1' = g_1$. This is true when the new elements falls between a pair of $(\times, O)$, with probability $\frac{l_1 + l_2 - g_0}{D}$.

- $g_1' = g_1 + 1, l_0' = l_0, l_2' = l_2, g_0' = g_0$, when the new elements falls between a pair of $(\times, \times)$, with probability $\frac{f-a-l_1-g_1}{D}$.

- $g_1' = g_1 + 1, l_2' = l_2 + 1, l_0' = l_0, g_0' = g_0$, when the new elements falls between a pair of $(O, \times)$, with probability $\frac{l_1}{D}$.

- $l_0' = l_0 - 1, l_2' = l_2 + 1, g_0' = g_0 + 1, g_1' = g_1$, when the new elements falls between a pair of $(O, O)$, with probability $\frac{l_0}{D}$.

- All values unchanged, when the "$-$" falls between other types of pairs, with probability $\frac{D-f+g_0+g_1}{D}$.

Denote $\Xi_D$ as the feasible set satisfying (10) with dimension $D \geq f$. Above reasoning builds a correspondence between $\Xi_D$ and $\Xi_{D+1}$. More precisely, we have

$$
\tilde{\mathcal{E}}_{D+1} = \sum_{\Xi_{D+1}} \left( \frac{l_0'}{f + g_0' + g_1'} + \frac{a(g_0' + l_2')}{(f + g_0' + g_1')f} \right) P_{a,f,D+1}(l_0', l_2', g_0', g_1')
$$

$$
= \sum_{\Xi_D} \left\{ \left( \frac{l_0}{f + g_0 + g_1 + 1} + \frac{a(g_0 + l_2 + 1)}{(f + g_0 + g_1 + 1)f} \right) \frac{l_1 + l_2 - g_0}{D} P_{a,f,D}(l_0, l_2, g_0, g_1) \right.
$$

$$
+ \left( \frac{l_0}{f + g_0 + g_1 + 1} + \frac{a(g_0 + l_2)}{(f + g_0 + g_1 + 1)f} \right) \frac{f - a - l_1 - g_1}{D} P_{a,f,D}(l_0, l_2, g_0, g_1)
$$

$$
+ \left( \frac{l_0}{f + g_0 + g_1 + 1} + \frac{a(g_0 + l_2 + 1)}{(f + g_0 + g_1 + 1)f} \right) \frac{l_1}{D} P_{a,f,D}(l_0, l_2, g_0, g_1)
$$

$$
+ \left( \frac{l_0 - 1}{f + g_0 + g_1 + 1} + \frac{a(g_0 + l_2 + 2)}{(f + g_0 + g_1 + 1)f} \right) \frac{l_0}{D} P_{a,f,D}(l_0, l_2, g_0, g_1)
$$

$$
\left. + \left( \frac{l_0}{f + g_0 + g_1} + \frac{a(g_0 + l_2)}{(f + g_0 + g_1)f} \right) \frac{D - f + g_0 + g_1}{D} P_{a,f,D}(l_0, l_2, g_0, g_1) \right\}.
$$

The increment can be computed as

$$
\tilde{\delta}_D \triangleq \tilde{\mathcal{E}}_{D+1} - \tilde{\mathcal{E}}_D
$$

$$
= \sum_{\Xi_D} \left\{ \frac{f - g_0 - g_1}{D} \left[ \left( \frac{l_0}{f + g_0 + g_1 + 1} - \frac{l_0}{f + g_0 + g_1} \right) + \left( \frac{a(g_0 + l_2 + 1)}{f + g_0 + g_1 + 1} - \frac{a(g_0 + l_2)}{f + g_0 + g_1} \right) \right] \right.
$$

$$
\left. - \frac{l_0}{D(f + g_0 + g_1 + 1)} - \frac{a(f - a - l_1 - g_1) - al_0}{Df(f + g_0 + g_1 + 1)} \right\} P_{a,f,D}(l_0, l_2, g_0, g_1)
$$

$$
= \sum_{\Xi_D} \left\{ \frac{(f - g_0 - g_1)[a(f + g_1 - l_2) - fl_0]}{Df(f + g_0 + g_1)(f + g_0 + g_1 + 1)} - \frac{(f - a)l_0 + a(f - a - l_1 - g_1)}{Df(f + g_0 + g_1 + 1)} \right\} P_{a,f,D}(l_0, l_2, g_0, g_1)
$$

$$
= \sum_{\Xi_D} \frac{2af(l_1 + g_1) - 2f(f - a)l_0 - 2a(f - a)(g_0 + g_1)}{Df(f + g_0 + g_1)(f + g_0 + g_1 + 1)} P_{a,f,D}(l_0, l_2, g_0, g_1)
$$

$$
= \mathbb{E}\left[ \frac{2af(l_1 + g_1) - 2f(f - a)l_0 - 2a(f - a)(g_0 + g_1)}{Df(f + g_0 + g_1)(f + g_0 + g_1 + 1)} \right]
$$

$$
= \mathbb{E}\left[ \frac{2af(f - a - h_1) - 2f(f - a)l_0 - 2a(f - a)(g_0 + g_1 + f - f)}{Df(f + g_0 + g_1)(f + g_0 + g_1 + 1)} \right]
$$

$$
= \mathbb{E}\left[ \frac{4a(f - a)}{D(f + g_0 + g_1)(f + g_0 + g_1 + 1)} \right] - \mathbb{E}\left[ \frac{2ah_1 + 2(f - a)l_0}{D(f + g_0 + g_1)(f + g_0 + g_1 + 1)} \right] - \mathbb{E}\left[ \frac{2a(f - a)}{Df(f + g_0 + g_1 + 1)} \right]
$$

$$
\triangleq 4a(f - a)E_0 - 2aE_1 - 2(f - a)E_2 - 2a(f - a)E_3, \tag{26}
$$

where

$$
E_0 = \mathbb{E}\left[ \frac{1}{D(f + g_0 + g_1)(f + g_0 + g_1 + 1)} \right], \quad E_1 = \mathbb{E}\left[ \frac{h_1}{D(f + g_0 + g_1)(f + g_0 + g_1 + 1)} \right],
$$

$$E_2 = \mathbb{E}\Big[\frac{l_0}{D(f+g_0+g_1)(f+g_0+g_1+1)}\Big], \quad E_3 = \mathbb{E}\Big[\frac{g_2}{Df(f+g_0+g_1+1)}\Big].$$

Note that here the expectations are taken w.r.t. the set size distribution with $a, f, D$. We can expand the terms of density function (25) to derive

$$P_{a,f,D}(l_0, l_2, g_0, g_1)$$

$$= \sum_{s=\max(0,D-2f+a)}^{D-f-1} \frac{(D-f-s)(D-f)!(f-a-1)!}{[D-(f+g_0+g_1)]![(f+g_0+g_1)-D+s]!g_1!(D-f-s-g_1)!}$$

$$\frac{(a-1)!}{(g_0+g_1-l_2)![D-s+l_2-(f+g_0+g_1)]!(f-a-l_1-g_1)!(f+g_1+l_1-D+s)!l_0!(a-l_0-1)!}$$

$$\frac{a!(f-a)!(D-f-1)!}{(D-1)!}.$$

Denote $a' = a-1$, $f' = f-1$, $D' = D-1$ and $l_0' = l_0 - 1$. We have

$$E_2 = \sum_{\Xi_D} \frac{l_0}{D(f+g_0+g_1)(f+g_0+g_1+1)} P_{a,f,D}(l_0, l_2, g_0, g_1)$$

$$= \sum_{\Xi_D} \frac{a(a-1)}{D-1} \cdot \frac{1}{D(f+g_0+g_1)(f+g_0+g_1+1)}$$

$$\sum_{s=\max(0,D'-2f'+a')}^{D'-f'-1} \frac{(D'-f'-s)(D'-f')!(f'-a'-1)!}{[D'-(f'+g_0+g_1)]![(f'+g_0+g_1)-D'+s]!g_1!(D'-f'-s-g_1)!}$$

$$\frac{(a'-1)!}{(g_0+g_1-l_2)![D'-s+l_2-(f'+g_0+g_1)]!(f'-a'-l_1-g_1)!(f'+g_1+l_1-D'+s)!l_0'!(a'-l_0'-1)!}$$

$$\frac{a'!(f'-a')!(D'-f'-1)!}{(D'-1)!}$$

$$= \sum_{\Xi_{D-1}} \frac{a(a-1)}{D-1} \frac{1}{D(f+g_0+g_1)(f+g_0+g_1+1)} P_{a-1,f-1,D-1}(l_0, l_2, g_0, g_1)$$

$$= \frac{a(a-1)}{D-1} \mathbb{E}_{a-1,f-1,D-1}\Big[\frac{1}{D(f+g_0+g_1)(f+g_0+g_1+1)}\Big]$$

$$\triangleq \frac{a(a-1)}{D-1} \bar{E}.$$

Here the subscript means that we are taking expectation w.r.t the set sizes when the number of "$O$", "$\times$" and "$-$" points is $(a-1, f-1, D-1)$. By symmetry, it can be shown similarly that

$$E_1 = \frac{(f-a)(f-a-1)}{D-1} \mathbb{E}_{a,f-1,D-1}\Big[\frac{1}{D(f+g_0+g_1)(f+g_0+g_1+1)}\Big] = \frac{(f-a)(f-a-1)}{D-1} \bar{E}.$$

Substituting above results into (26), we obtain

$$\tilde{\delta}_D = 2a(f-a)\Big[2E_0 - \frac{f-2}{D-1}\bar{E} - E_3\Big].$$

To compute $E_0$, note that with $a$, $f$ and $D$, variable $g_2$ is distributed as $\text{hyper}(D-1, D-f, D-f-1)$. For $\bar{E}$, the distribution becomes $\text{hyper}(D-2, D-f, D-f-1)$. Since $f+g_0+g_1 = D-g_2$, we deduce

$$E_0 = \sum_{s=\max(0,D-2f)}^{D-f-1} \frac{1}{D(D-s)(D-s+1)} \frac{\binom{D-f-1}{s}\binom{f}{D-f-s}}{\binom{D-1}{D-f}}$$

$$= \sum_{s=\max(0,D-2f)}^{D-f-1} \frac{1}{D(D-s)(D-s+1)} \frac{(D-f-1)!f!}{s!(D-f-s-1)!(D-f-s)!(-D+2f+s)!} \frac{(D-f)!(f-1)!}{(D-1)!},$$

and

$$\bar{E} = \sum_{s=\max(0,D-2f+1)}^{D-f-1} \frac{1}{D(D-s)(D-s+1)} \frac{\binom{D-f-1}{s}\binom{f-1}{D-f-s}}{\binom{D-2}{D-f}}$$

$$= \sum_{s=\max(0,D-2f+1)}^{D-f-1} \frac{1}{D(D-s)(D-s+1)} \frac{(D-f)!(f-2)!}{(D-2)!} \cdot$$

$$\frac{(D-f-1)!(f-1)!}{s!(D-f-s-1)!(D-f-s)!(-D+2f+s-1)!}.$$

For $\forall D \geq f$, we have

$$\frac{f-2}{D-1}\bar{E} \leq \sum_{s=\max(0,D-2f)}^{D-f-1} \frac{(f-2)(D-1)(-D+2f+s)}{D(D-1)f(f-1)(D-s)(D-s+1)}$$

$$\frac{(D-f-1)!f!}{s!(D-f-s-1)!(D-f-s)!(-D+2f+s)!} \frac{(D-f)!(f-1)!}{(D-1)!}$$

$$\leq \mathbb{E}\Big[\frac{(f-2)(f-(D-f-g_2))}{Df(f-1)(D-g_2)(D-g_2+1)}\Big]$$

$$< \mathbb{E}\Big[\frac{(f-g_0-g_1)}{Df(f+g_0+g_1)(f+g_0+g_1+1)}\Big].$$

Consequently, we have

$$\tilde{\delta}_D > 2a(f-a)\mathbb{E}\Big[\frac{2}{D(f+g_0+g_1)(f+g_0+g_1+1)} - \frac{f-g_0-g_1}{Df(f+g_0+g_1)(f+g_0+g_1+1)}$$

$$- \frac{f+g_0+g_1}{Df(f+g_0+g_1)(f+g_0+g_1+1)}\Big]$$

$$= 0,$$

and note that this holds for $\forall D \geq K$. The proof is complete. $\qquad\square$

## A.6 Proof of Theorem 3.5

**Theorem 3.5** (Uniform Superiority). *For any two binary vectors $\boldsymbol{v}, \boldsymbol{w} \in \{0,1\}^D$ with $J \neq 0$ or $1$, it holds that $Var[\hat{J}_{\sigma,\pi}(\boldsymbol{v}, \boldsymbol{w})] < Var[\hat{J}_{MH}(\boldsymbol{v}, \boldsymbol{w})]$.*

*Proof.* By assumption we have $0 < a < f$. To compare $Var[\hat{J}_{\sigma,\pi}]$ with $Var[\hat{J}_{MH}] = \frac{J(1-J)}{K} = \frac{J}{K} + \frac{(K-1)J^2}{K} - J^2$, it suffices to compare $\tilde{\mathcal{E}}$ with $J^2$. When $D = f$, we know that the location vector $\boldsymbol{x}$ of $(\boldsymbol{v}, \boldsymbol{w})$ contains no "$-$" elements. It is easy to verify that in this case, $|\mathcal{G}_0| = |\mathcal{G}_1| = |\mathcal{L}_2| = 0$, and $|\mathcal{L}_0|$ follows hyper$(f-1, a, a-1)$. By Theorem 3.2, it follows that when $D = f$,

$$\tilde{\mathcal{E}}_D = \frac{1}{f}\mathbb{E}[|\mathcal{L}_0|] = \frac{a(a-1)}{f(f-1)} = J\tilde{J} < J^2.$$

Recall the definition $\tilde{J} = \frac{a-1}{f-1}$, which is always less than $J$. On the other hand, as $D \to \infty$, we have $|\mathcal{L}_0| \to 0$, $|\mathcal{L}_2| \to a$, $|\mathcal{G}_0| \to a$ and $|\mathcal{G}_1| \to f - a$. We can easily show that

$$\tilde{\mathcal{E}}_D \to J^2, \quad \text{as } D \to \infty.$$

By Lemma 3.4, the sequence $(\tilde{\mathcal{E}}_f, \tilde{\mathcal{E}}_{f+1}, \tilde{\mathcal{E}}_{f+2}, ...)$ is strictly increasing. Since it is convergent with limit $J^2$, by the Monotone Convergence Theorem we know that $\tilde{\mathcal{E}}_D < J^2, \forall D \geq f$. $\qquad\square$

## A.7 Proof of Proposition 3.6

**Proposition 3.6** (Consistent Improvement). *Suppose $f$ is fixed. In terms of $a$, the variance ratio $\rho(a) = \frac{Var[\hat{J}_{MH}(\boldsymbol{v}, \boldsymbol{w})]}{Var[\hat{J}_{\sigma,\pi}(\boldsymbol{v}, \boldsymbol{w})]}$ is constant for any $0 < a < f$.*

*Proof.* Let $\tilde{\mathcal{E}}$ be defined as in Theorem 3.2. Assume that $D$ and $f$ are fixed and $a$ is variable. Firstly, we can write the variance ratio explicitly as

$$\rho(a) = \frac{\frac{J-J^2}{K}}{\frac{J}{K} + \frac{(K+1)\tilde{\mathcal{E}}}{K} - J^2} = \frac{1 - J}{1 - J - (K-1)(J - \frac{\tilde{\mathcal{E}}}{J})}.$$

We now show that the term $J - \frac{\tilde{\mathcal{E}}}{J} = C(1 - J)$, where $C$ is some constant independent of $J$ (i.e., $a$). Then, for fixed $D$ and $f$, by cancellation $\rho(a)$ would be constant for all $0 < a < f$. We have

$$\begin{aligned}
J - \frac{\tilde{\mathcal{E}}}{J} &= \frac{a}{f} - \mathbb{E}_{a,f,D}\left[\frac{fl_0}{a(f + g_0 + g_1)} + \frac{g_0 + l_2}{f + g_0 + g_1}\right] \\
&= \mathbb{E}\left[\frac{a^2(f + g_0 + g_1) - f^2l_0 - af(g_0 + l_2)}{af(f + g_0 + g_1)}\right] \\
&= \mathbb{E}\left[\frac{a(a - f)(g_0 + g_1) + a^2f + afg_1 - f^2l_0 - afl_2}{af(f + g_0 + g_1)}\right] \\
&= \mathbb{E}\left[\frac{a(a - f)(g_0 + g_1) + af(l_0 + l_1) + afg_1 - f^2l_0}{af(f + g_0 + g_1)}\right] \\
&= \mathbb{E}\left[\frac{a(a - f)(g_0 + g_1) + f(a - f)l_0 + af(f - a - h_1)}{af(f + g_0 + g_1)}\right],
\end{aligned} \qquad (27)$$

where we use the constraints (10) that $l_0 + l_1 + l_2 = a$ and $l_1 + g_1 + h_1 = f - a$. We now study the three terms respectively. We have

$$\mathbb{E}\left[\frac{a(a - f)(g_0 + g_1)}{af(f + g_0 + g_1)}\right] = -(1 - J)\mathbb{E}\left[\frac{g_0 + g_1}{f + g_0 + g_1}\right] \triangleq -E'(1 - J).$$

We have shown in the proof of Lemma 3.4 that

$$\mathbb{E}_{a,f,D}\left[\frac{l_0}{f + g_0 + g_1}\right] = \frac{a(a - 1)}{D - 1}\mathbb{E}_{a-1,f-1,D-1}\left[\frac{1}{f + g_0 + g_1}\right] \triangleq \frac{a(a - 1)}{D - 1}E^*,$$

and by symmetry it holds that

$$\mathbb{E}_{a,f,D}\left[\frac{h_1}{f + g_0 + g_1}\right] = \frac{(f - a)(f - a - 1)}{D - 1}E^*.$$

Note that Since $f$ is fixed, $(|\mathcal{G}_0| + |\mathcal{G}_1|)$ is distributed independent of $a$. Consequently, $E'$ and $E^*$ are both independent of $a$. Next, we obtain

$$\mathbb{E}\left[\frac{f(a - f)l_0}{af(f + g_0 + g_1)}\right] = -(1 - J)\frac{f(a - 1)}{D - 1}E^*,$$

and

$$\mathbb{E}\left[\frac{af(f - a - h_1)}{af(f + g_0 + g_1)}\right] = (1 - J)fE^* - (1 - J)\frac{f(f - a - 1)}{D - 1}E^*.$$

Summing up the terms and substituting into (27), we derive

$$J - \frac{\tilde{\mathcal{E}}}{J} = C(1 - J),$$

where $C = -E' + (f - \frac{f(f-2)}{D-1})E^*$, which is independent of $a$. Taking into $\rho(a)$, we get

$$\rho(a) = \frac{1 - J}{1 - J - (K - 1)C(1 - J)} = \frac{1}{1 - (K - 1)C},$$

which is a constant only depending on $f$, $D$ and $K$. This completes the proof. $\qquad\square$

A.8 Proof of Theorem 5.1

*Proof.* Denote $\mathbb{E}[\mathbb{1}_k] := \mathbb{E}[\mathbb{1}\{h_k(\boldsymbol{v}) = h_k(\boldsymbol{w})\}]$ for any $1 \leq k \leq K$. We first recall some notations. We have $\boldsymbol{v}, \boldsymbol{w} \in \{0, 1\}^D$, and $a$ and $f$ are defined in (5). Denote $\mathcal{B}_1 = \{i : \boldsymbol{x}_i = O\}$, $\mathcal{B}_2 = \{i : \boldsymbol{x}_i = \times\}$ and $\mathcal{B}_3 = \{i : \boldsymbol{x}_i = -\}$ as the sets of three types of points, respectively. For $a \leq j \leq D$ and $1 \leq k \leq K$, define

$$\mathcal{A}_-(j) = \{\boldsymbol{x}_i : (i + k - 1 \ mod \ D) + 1 \leq j\},$$
$$\mathcal{A}_+(j) = \{\boldsymbol{x}_i : (i + k - 1 \ mod \ D) + 1 > j\}.$$

Let $n_{-,1}(j) = |\{\boldsymbol{x}_i = O : i \in \mathcal{A}_-(j)\}|$ be the number of "$O$" points in $\mathcal{A}_-(j)$. Analogously let $n_{-,2}(j), n_{-,3}(j)$ be the number of "$\times$" and "$-$" points in $\mathcal{A}_-(j)$, and $n_{+,1}(j), n_{+,2}(j), n_{+,3}(j)$ be the number of "$O$", "$\times$" and "$-$" points in $\mathcal{A}_+(j)$. For any $i$, denote $i^* = (i + k - 1 \ mod \ D) + 1$, $i^\# = (i - k - 1 \ mod \ D) + 1$.

Our analysis starts with the decomposition of hash collision probability,

$$\mathbb{E}[\mathbb{1}_k] = P\Big[h_k(\boldsymbol{v}) = h_k(\boldsymbol{w})\Big] = \sum_{j=1}^{D} P\Big[h_k(\boldsymbol{v}) = h_k(\boldsymbol{w}) = j\Big], \tag{28}$$

where recall $h(\cdot)$ is the hash sample. Consider the process for generating the hash. As before, we look at the location vector $\boldsymbol{x}$. In Method 2, we first permute $\boldsymbol{x}$ by $\pi$ to get $\pi(\boldsymbol{x})$. Then the $k$-th hash samples collide if the minimum of $\pi_{\rightarrow k}(\pi(\boldsymbol{x}))$ is "$O$". One key observation is that, when applying $\pi_{\rightarrow k}$, the random index for the $i$-th element in $\pi(\boldsymbol{x})$ is exactly the one used for $\boldsymbol{x}_{i^\#}$ (shifted backwards) in the initial permutation. A toy example in provided in Figure 8 to help understand the reasoning.

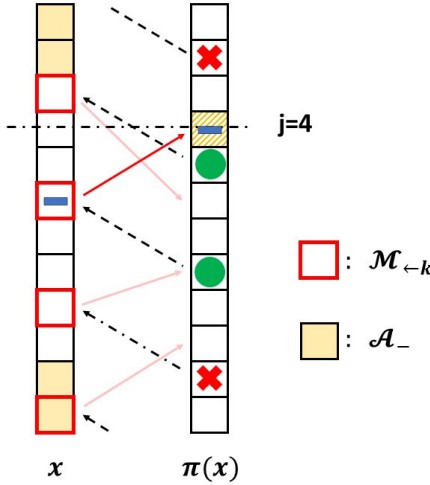

Figure 8: Illustration of C-MinHash-$(\pi, \pi)$ hash collision, with $k = 2$. Here, "circulant right" means "circulant down". Small indices correspond to upper elements.

Further denote set $\mathcal{M} = \{\pi(i) : \pi(i) \notin \mathcal{B}_3\}$ be the collection of indices of initially permuted vector $\pi(\boldsymbol{x})$ that are not "$-$" points, and $\mathcal{M}_{\leftarrow k} = (\mathcal{M} - k - 1 \ mod \ D) + 1$ be the corresponding indices shifted backwards. In Figure 8, $\mathcal{M} = \{2, 5, 8, 11\}$. Also, $\pi(6) = 4$, and the permutation maps are described by the red arrows. Consequently, in $\pi_{\rightarrow k}(\pi(\boldsymbol{x}))$, the 8-th element ("$O$") in $\pi(\boldsymbol{x})$ will be permuted to the same index of the $8 - 2 = 6$-th element in $\boldsymbol{x}$, which equals to 4. It is important to notice that, when considering the $k$-th collision, only points in $\mathcal{M}_{\leftarrow k}$ matters. Hence, we deduct:

- **(Collision condition)** Denote $i = \arg\min_{t \in \mathcal{M}_{\leftarrow k}} \pi(\boldsymbol{x}_t)$ be the location of minimal permutation index in $\mathcal{M}_{\leftarrow k}$. The $k$-th collision occurs at $j$ i.f.f. $\pi(i) = j$, and the $i^*$-th element in $\pi(\boldsymbol{x})$ must be a "$O$" point. Recall the definition $i^* = (i + k - 1 \ mod \ D) + 1$.

In Figure 8, consider $i = 6$ and $j = 4$ for example. Above condition means that the $i = 6$-th element in $\boldsymbol{x}$ ("$-$" in red bold border) is permuted to the $j = 4$-th position, and it is above all other permuted elements with red bold borders. Meanwhile, the $i + k = 6 + 2 = 8$-th element in $\pi(\boldsymbol{x})$ must be a "$O$". Figure 8 exactly satisfies the condition, so it depicts a collision. Mathematically, we have

$$
\begin{aligned}
\mathbb{E}[\mathbb{1}_k] &= \sum_{j=1}^{D} P\Big[h_k(\boldsymbol{v}) = h_k(\boldsymbol{w}) = j\Big] \\
&= \sum_{j=1}^{D} \sum_{i=1}^{D} P\Big[\pi(i) = j, \pi^{-1}(i^*) \in \mathcal{B}_1\Big],
\end{aligned}
\tag{29}
$$

where $i = \arg\min_{t \in \mathcal{M}_{\leftarrow k}} \pi(\boldsymbol{x}_t)$. In this expression, everything is random of $\pi$, except for the set $\mathcal{B}_1$ which is fixed given the data.

Now we will focus on deriving the probability for a fixed $i$ and $j$ in (29). Our analysis will be conditional on the collection of variables $Z$ which is defined as follows. Let $z_{-,1}$, $z_{-,2}$ and $z_{-,3}$ be the number of "$O$", "$\times$" and "$-$" points in $\mathcal{A}_-(j) \cap \mathcal{M}_{\leftarrow k}^c$, and $z_{+,1}$, $z_{+,2}$ and $z_{+,3}$ be the number of "$O$", "$\times$" and "$-$" points in $\mathcal{A}_+(j) \cap \mathcal{M}_{\leftarrow k}^c$, respectively. Here $\mathcal{M}_{\leftarrow k}^c$ represents the complement of $\mathcal{M}_{\leftarrow k}$. Notice that $Z$ (and its density function) depends on different $j$ since $\mathcal{A}_-(j)$ and $\mathcal{A}_+(j)$ depends on $j$. For the ease of notation we suppress the information of $j$ in $Z$ (and $z$'s). It is easy to see that $Z = (z_{-,k}|_1^3, z_{+,k}|_1^3)$ follows $\text{hyper}(D, D - f, n_{-,k}(j)|_1^3, n_{+,k}(j)|_1^3)$. Denote the domain of $Z$ and $\Theta_j$. Conditional on $Z$, we obtain

$$
\begin{aligned}
\mathbb{E}[\mathbb{1}_k] &= \sum_{j=1}^{D} \sum_{Z \in \Theta_j} \sum_{i=1}^{D} P\Big[\pi(i) = j, \pi^{-1}(i^*) \in \mathcal{B}_1|Z\Big] P_j\Big[Z\Big] \\
&= \sum_{j=1}^{D} \sum_{Z \in \Theta_j} \sum_{i=1}^{D} P\Big[\pi^{-1}(i^*) \in \mathcal{B}_1|\pi(i) = j, Z\Big] P\Big[\pi(i) = j|Z\Big] P_j\Big[Z\Big] \\
&\triangleq \sum_{j=1}^{D} \sum_{Z \in \Theta_j} \sum_{i=1}^{D} \Gamma(i,j) P_j\Big[Z\Big],
\end{aligned}
\tag{30}
$$

with $i = \arg\min_{t \in \mathcal{M}_{\leftarrow k}} \pi(\boldsymbol{x}_t)$. We will carefully compute the probabilities in the summation. Basically, the key is that elements in $\mathcal{M}$ need to be controlled, i.e. smaller than $j$, and other positions can be arbitrary. Given $Z$, this means that we need to put $r_1 = a - z_{-,1} - z_{+,1}$ type "$O$" points, $r_2 = f - a - z_{-,2} - z_{+,2}$ type "$\times$" points and $r_3 = D - f - z_{-,3} - z_{+,3}$ type "$-$" points no smaller than $j$, with $\pi(i) = j$ exactly. Also note that there are fixed $b_0 = \sum_{k=1}^{3} z_{+,k}$ type "$-$" points no smaller than $j$.

With all these definitions and reasoning, we are ready to proceed with the proof. Based on $\boldsymbol{x}_i$, we have three general cases.

**1) $\boldsymbol{x}_i \in \mathcal{B}_1$.** The first case is that $\boldsymbol{x}_i = O$.

**Case 1a) $j < i^*$.** Firstly, we consider the case where $j < i^*$. By combinatorial theory we have

$$
\begin{aligned}
P\Big[\pi(i) = j|Z\Big] &= P\Big[\pi(i) = j|\pi^{-1}(j) \notin \mathcal{B}_3, Z\Big] P\Big[\pi^{-1}(j) \notin \mathcal{B}_3|Z\Big] \\
&= \frac{1}{r_1 + r_2} \frac{\binom{b_0}{r_3}\binom{D-j-b_0}{r_1+r_2-1}}{\binom{D-f}{r_3}\binom{f}{r_1+r_2}} P\Big[\pi^{-1}(j) \notin \mathcal{B}_3|Z\Big] \\
&\triangleq \tilde{P}_1 \cdot P\Big[\pi^{-1}(j) \notin \mathcal{B}_3|Z\Big],
\end{aligned}
\tag{31}
$$

where the second probability is that the $j$-th element in $\pi(\boldsymbol{x})$ is not "$-$". Conditional on $Z$, the probability is dependent on $j^{\#}$:

$$
P\Big[\pi^{-1}(j) \notin B_3|Z\Big] = \sum_{p=1}^{3} \mathbb{1}\{j^{\#} \in \mathcal{B}_p\}(1 - \frac{z_{-,p}}{n_{-,p}(j)}).
$$

Combining with (31) we obtain

$$P\Big[\pi(i) = j|Z\Big] = \tilde{P}_1 \sum_{p=1}^{3} \mathbb{1}\{j^{\#} \in \mathcal{B}_p\}(1 - \frac{z_{-,p}}{n_{-,p}(j)}). \tag{32}$$

Next we compute $P[\pi^{-1}(i^*) \in \mathcal{B}_1|\pi(i) = j, Z]$. Note that given the conditions, $\pi^{-1}(i^*)$ has two cases: 1) it comes from $\mathcal{M}_{\leftarrow k}$ (i.e. it is one of the elements with red bold border); 2) Otherwise. We then can write

$$P\Big[\pi^{-1}(i^*) \in \mathcal{B}_1|\pi(i) = j, Z\Big]$$
$$= P\Big[\pi^{-1}(i^*) \in \mathcal{B}_1|\pi^{-1}(i^*) \in \mathcal{M}_{\leftarrow k}, \pi(i) = j, Z\Big] P\Big[\pi^{-1}(i^*) \in \mathcal{M}_{\leftarrow k}|\pi(i) = j, Z\Big]$$
$$\quad + P\Big[\pi^{-1}(i^*) \in \mathcal{B}_1|\pi^{-1}(i^*) \notin \mathcal{M}_{\leftarrow k}, \pi(i) = j, Z\Big] P\Big[\pi^{-1}(i^*) \notin \mathcal{M}_{\leftarrow k}|\pi(i) = j, Z\Big]$$
$$= (1 - \frac{z_{+,1}}{n_{+,1}(j)}) \Big[ \frac{r_1 + r_2 - 1}{D - j - b_0} \frac{r_1 - 1}{r_1 + r_2 - 1} + (1 - \frac{r_1 + r_2 - 1}{D - j - b_0}) \frac{a - r_1}{f - r1 - r2} \Big]$$
$$\triangleq (1 - \frac{z_{+,1}}{n_{+,1}(j)}) \Big[ \frac{r_1 - 1}{D - j - b_0} + (1 - \frac{r_1 + r_2 - 1}{D - j - b_0})J^* \Big]$$
$$\triangleq (1 - \frac{z_{+,1}}{n_{+,1}(j)})\bar{J}_1. \tag{33}$$

Combining (32) and (33), we obtain when $i \in \mathcal{B}_1$ and $j < i^*$,

$$\Gamma(i,j) = \sum_{p=1}^{3} \mathbb{1}\{j^{\#} \in \mathcal{B}_p\}(1 - \frac{z_{-,p}}{n_{-,p}(j)})(1 - \frac{z_{+,1}}{n_{+,1}(j)})\tilde{P}_1 \bar{J}_1. \tag{34}$$

**Case 1b)** $j = i^*$. Similarly approach also applies to the situation with $j = i^*$. In this case,

$$P\Big[\pi^{-1}(j) \notin \mathcal{B}_3|Z\Big] = (1 - \frac{z_{-,1}}{n_{-,1}(j)})\tilde{P}_1, \qquad P[\pi^{-1}(i^*) \in \mathcal{B}_1|\pi(i) = j, Z] = 1. \tag{35}$$

The equations are because $(i^*)^{\#} = i \in \mathcal{B}_1$, and equivalently, $\pi^{-1}(i^*) = \pi^{-1}(j) = i \in \mathcal{B}_1$.

**Case 1c)** $j > i^*$. I this case, we still have

$$P\Big[\pi(i) = j|Z\Big] = \tilde{P}_1 \sum_{p=1}^{3} \mathbb{1}\{j^{\#} \in \mathcal{B}_p\}(1 - \frac{z_{-,p}}{n_{-,p}(j)}),$$

but the probability of $\pi_{-1}(i^*)$ being "$O$" is different. Since $j > i^*$, this event now depends on $z_{-,p}$, $p = 1, 2, 3$. More specifically,

$$P[\pi^{-1}(i^*) \in \mathcal{B}_1|\pi(i) = j, Z]$$
$$= \Big[ \mathbb{1}\{j^{\#} \in \mathcal{B}_1\}(1 - \frac{z_{-,1}}{n_{-,1}(j) - 1}) + \sum_{p=2,3} \mathbb{1}\{j^{\#} \in \mathcal{B}_p\}(1 - \frac{z_{-,p}}{n_{-,p}(j)}) \Big] J^*.$$

Therefore, when $i \in \mathcal{B}_1$ and $j > j^*$, it holds that

$$\Gamma(i,j) = (1 - \frac{z_{-,1}}{n_{-,1}(j)}) \Big[ \mathbb{1}\{j^{\#} \in \mathcal{B}_1\}(1 - \frac{z_{-,1}}{n_{-,1}(j) - 1}) + \sum_{p=2,3} \mathbb{1}\{j^{\#} \in \mathcal{B}_p\}(1 - \frac{z_{-,p}}{n_{-,p}(j)}) \Big] J^*. \tag{36}$$

**2) $x_i \in \mathcal{B}_2$.** The case where $x_i \in \mathcal{B}_2$ can be analyzed using similar arguments. For conciseness, we mainly present the final results.

**Case 2a)** $j < i^*$. The calculation ends up in the same form. We have

$$P\Big[\pi(i) = j|Z\Big] = \tilde{P}_2 \sum_{p=1}^{3} \mathbb{1}\{j^{\#} \in \mathcal{B}_p\}(1 - \frac{z_{-,p}}{n_{-,p}(j)}),$$

with $\tilde{P}_2 = \tilde{P}_1$. In addition,

$$P\left[\pi^{-1}(i^*) \in \mathcal{B}_1 | \pi(i) = j, Z\right] = (1 - \frac{z_{+,2}}{n_{+,2}(j)})\bar{J}_2,$$

where $\bar{J}_2 = \frac{r_1}{D-j-b_0} + (1 - \frac{r_1+r_2-1}{D-j-b_0})J^*$. Hence, when $x_i \in \mathcal{B}_2$ and $j < i^*$, we have

$$\Gamma(i,j) = \sum_{p=1}^{3} \mathbb{1}\{j^\# \in \mathcal{B}_p\}(1 - \frac{z_{-,p}}{n_{-,p}(j)})(1 - \frac{z_{+,2}}{n_{+,2}(j)})\tilde{P}_2\bar{J}_2. \tag{37}$$

**Case 2b)** $j = i^*$. In this case, $\Gamma(i,j)$ simply equals to 0, since $\pi^{-1}(i^*) = \pi^{-1}(j) \in \mathcal{B}_2$. The probability of $\pi^{-1}(i^*)$ being "$O$" is 0.

**Case 2c)** $j > i^*$. Omitting the details, we have

$$\Gamma(i,j) = (1 - \frac{z_{-,2}}{n_{-,2}(j)})\left[\mathbb{1}\{j^\# \in \mathcal{B}_2\}(1 - \frac{z_{-,2}}{n_{-,2}(j)-1}) + \sum_{p=1,3}\mathbb{1}\{j^\# \in \mathcal{B}_p\}(1 - \frac{z_{-,p}}{n_{-,p}(j)})\right]J^*. \tag{38}$$

**2) $x_i \in \mathcal{B}_3$.**

**Case 3a)** $j < i^*$. The expression is different from previous two, in that we need $\pi^{-1}(j) \in B_3$.

$$P\left[\pi(i) = j | Z\right] = P\left[\pi(i) = j | \pi^{-1}(j) \in B_3, Z\right]P\left[\pi^{-1}(j) \in B_3 | Z\right]$$

$$= \tilde{P}_3\sum_{p=1}^{3}\mathbb{1}\{j^\# \in \mathcal{B}_p\}\frac{z_{-,p}}{n_{-,p}(j)},$$

where

$$\tilde{P}_3 = \frac{1}{r_3}\frac{\binom{b_0}{r_3-1}\binom{D-j-b_0}{r_1+r_2}}{\binom{D-f}{r_3}\binom{f}{r_1+r_2}}.$$

Moreover, we have

$$P\left[\pi^{-1}(i^*) \in \mathcal{B}_1 | \pi(i) = j, Z\right] = (1 - \frac{z_{+,3}}{n_{+,3}(j)})\bar{J}_3,$$

with $\bar{J}_3 = \frac{r_1}{D-j-b_0} + (1 - \frac{r_1+r_2}{D-j-b_0})J^*$. Combining parts together we obtain

$$\Gamma(i,j) = \sum_{p=1}^{3}\mathbb{1}\{j^\# \in \mathcal{B}_p\}\frac{z_{-,p}}{n_{-,p}(j)}(1 - \frac{z_{+,3}}{n_{+,3}(j)})\tilde{P}_3\bar{J}_3. \tag{39}$$

**Case 3b)** $j = i^*$. By same reasoning as Case 2a), $\Gamma(i,j) = 0$.

**Case 3c)** $j > i^*$. We have in this case

$$\Gamma(i,j) = \left[\mathbb{1}\{j^\# \in \mathcal{B}_3\}\frac{z_{-,3}}{n_{-,3}(j)}\frac{n_{-,3}(j)-z_{-,3}}{n_{-,3}(j)-1} + (1 - \frac{z_{-,3}}{n_{-,3}(j)})\sum_{p=1,2}\mathbb{1}\{j^\# \in \mathcal{B}_p\}\frac{z_{-,p}}{n_{-,p}(j)}\right]J^*$$

$$= \sum_{p=1}^{3}\mathbb{1}\{j^\# \in \mathcal{B}_p\}\frac{z_{-,p}}{n_{-,p}(j)}(1 - \frac{z_{-,3} - \mathbb{1}\{p=3\}}{n_{-,3}(j)} - \mathbb{1}\{p=3\})J^*. \tag{40}$$

Finally, combining (30), (34), (35), (36), (37), (38), (39) and (40) and re-organizing terms, the proof is complete.

$\square$

## B    MORE NUMERICAL JUSTIFICATION ON C-MINHASH-$(\pi, \pi)$

The "Words" dataset (Li & Church, 2005) (which is publicly available) contains a large number of word vectors, with the $i$-th entry indicating whether this word appears in the $i$-th document, for a total of $D = 2^{16}$ documents. The key statistics of the 120 selected word pairs are presented in Table 1. Those 120 pairs of words are more or less randomly selected except that we make sure they cover a wide spectrum of data distributions. Denote $d$ as the number of non-zero entries in the vector. Table 1 reports the density $\tilde{d} = d/D$ for each word vector, ranging from 0.0006 to 0.6. The Jaccard similarity $J$ ranges from 0.002 to 0.95.

In Figures 9 - 16, we plot the empirical MSE along with the empirical bias$^2$ for $\hat{J}_{\pi,\pi}$, as well as the empirical MSE for $\hat{J}_{\sigma,\pi}$. Note that for $D$ this large, it is numerically difficult to evaluate the theoretical variance formulas. From the results in the Figures, we can observe

- For all the data pairs, the MSE of C-MinHash-$(\pi, \pi)$ estimator overlaps with the empirical MSE of C-MinHash-$(\sigma, \pi)$ estimator for all $K$ from 1 up to 4096.
- The bias$^2$ is several orders of magnitudes smaller than the MSE, in all data pairs. This verifies that the bias of $\hat{J}_{\pi,\pi}$ is extremely small in practice and can be safely neglected.

We have many more plots on more data pairs. Nevertheless, we believe the current set of experiments on this "Words" dataset should be sufficient to verify that, the proposed C-MinHash-$(\pi, \pi)$ could give indistinguishable Jaccard estimation accuracy in practice compared with C-MinHash-$(\sigma, \pi)$.

Table 1: 120 selected word pairs from the *Words* dataset (Li & Church, 2005). For each pair, we report the density $\tilde{d}$ (number of non-zero entries divided by $D = 2^{16}$) for each word as well as the Jaccard similarity $J$. Both $\tilde{d}$ and $J$ cover a wide range of values.

| | $\tilde{d}_1$ | $\tilde{d}_2$ | $J$ | | $\tilde{d}_1$ | $\tilde{d}_2$ | $J$ |
|---|---|---|---|---|---|---|---|
| ABOUT - INTO | 0.302 | 0.125 | 0.258 | NEW - WEB | 0.291 | 0.194 | 0.224 |
| ABOUT - LIKE | 0.302 | 0.140 | 0.281 | NEWS - LIKE | 0.168 | 0.140 | 0.172 |
| ACTUAL - DEVELOPED | 0.017 | 0.030 | 0.071 | NO - WELL | 0.220 | 0.120 | 0.244 |
| ACTUAL - GRABBED | 0.017 | 0.002 | 0.016 | NOT - IT | 0.281 | 0.295 | 0.437 |
| AFTER - OR | 0.103 | 0.356 | 0.220 | NOTORIOUSLY - LOCK | 0.0006 | 0.006 | 0.004 |
| AND - PROBLEM | 0.554 | 0.044 | 0.070 | OF - THEN | 0.570 | 0.104 | 0.168 |
| AS - NAME | 0.280 | 0.144 | 0.204 | OF - WE | 0.570 | 0.226 | 0.361 |
| AT - CUT | 0.374 | 0.242 | 0.052 | OPPORTUNITY - COUNTRIES | 0.029 | 0.024 | 0.066 |
| BE - ONE | 0.323 | 0.221 | 0.403 | OUR - THAN | 0.244 | 0.125 | 0.245 |
| BEST - AND | 0.136 | 0.554 | 0.228 | OVER - BACK | 0.148 | 0.160 | 0.233 |
| BRAZIL - OH | 0.010 | 0.031 | 0.019 | OVER - TWO | 0.148 | 0.121 | 0.289 |
| BUT - MANY | 0.167 | 0.116 | 0.340 | PEAK - SHOWS | 0.006 | 0.033 | 0.026 |
| CALLED - BUSINESSES | 0.016 | 0.018 | 0.043 | PEOPLE - BY | 0.121 | 0.425 | 0.228 |
| CALORIES - MICROSOFT | 0.002 | 0.045 | 0.0003 | PEOPLE - INFO | 0.121 | 0.138 | 0.117 |
| CAN - FROM | 0.243 | 0.326 | 0.444 | PICKS - BOOST | 0.007 | 0.005 | 0.007 |
| CAN - SEARCH | 0.243 | 0.214 | 0.237 | PLANET - REWARD | 0.013 | 0.003 | 0.018 |
| COMMITTED - PRODUCTIVE | 0.013 | 0.004 | 0.029 | PLEASE - MAKE | 0.168 | 0.141 | 0.195 |
| CONTEMPORARY - FLASH | 0.011 | 0.021 | 0.013 | PREFER - PUEDE | 0.010 | 0.003 | 0.0001 |
| CONVENIENTLY - INDUSTRIES | 0.003 | 0.011 | 0.009 | PRIVACY - FOUND | 0.126 | 0.136 | 0.053 |
| COPYRIGHT - AN | 0.218 | 0.290 | 0.209 | PROSECUTION - MAXIMIZE | 0.002 | 0.003 | 0.006 |
| CREDIT - CARD | 0.046 | 0.041 | 0.285 | RECENTLY - INT | 0.028 | 0.007 | 0.014 |
| DE - WEB | 0.117 | 0.194 | 0.091 | REPLY - ACHIEVE | 0.013 | 0.012 | 0.023 |
| DO - GOOD | 0.174 | 0.102 | 0.276 | RESERVED - BEEN | 0.172 | 0.141 | 0.108 |
| EARTH - GROUPS | 0.021 | 0.035 | 0.056 | RIGHTS - FIND | 0.187 | 0.144 | 0.166 |
| EXPRESSED - FRUSTRATED | 0.010 | 0.002 | 0.024 | RIGHTS - RESERVED | 0.187 | 0.172 | 0.877 |
| FIND - HAS | 0.144 | 0.228 | 0.214 | SCENE - ABOUT | 0.012 | 0.301 | 0.029 |
| FIND - SITE | 0.144 | 0.275 | 0.212 | SEE - ALSO | 0.138 | 0.166 | 0.291 |
| FIXED - SPECIFIC | 0.011 | 0.039 | 0.054 | SEIZE - ANYTHING | 0.0007 | 0.037 | 0.012 |
| FLIGHT - TRANSPORTATION | 0.011 | 0.018 | 0.040 | SHOULDERS - GORGEOUS | 0.003 | 0.004 | 0.028 |
| FOUND - DE | 0.136 | 0.117 | 0.039 | SICK - FELL | 0.008 | 0.008 | 0.085 |
| FRANCISCO - SAN | 0.025 | 0.049 | 0.476 | SITE - CELLULAR | 0.275 | 0.006 | 0.010 |
| GOOD - BACK | 0.102 | 0.160 | 0.220 | SOLD - LIVE | 0.018 | 0.064 | 0.055 |
| GROUPS - ORDERED | 0.035 | 0.011 | 0.034 | SOLO - CLAIMS | 0.010 | 0.012 | 0.007 |
| HAPPY - CONCEPT | 0.029 | 0.013 | 0.054 | SOON - ADVANCE | 0.040 | 0.017 | 0.057 |
| HAVE - FIRST | 0.267 | 0.151 | 0.320 | SPECIALIZES - ACTUAL | 0.003 | 0.017 | 0.008 |
| HAVE - US | 0.267 | 0.284 | 0.349 | STATE - OF | 0.101 | 0.570 | 0.165 |
| HILL - ASSURED | 0.020 | 0.004 | 0.011 | STATES - UNITED | 0.061 | 0.062 | 0.591 |
| HOME - SYNTHESIS | 0.365 | 0.002 | 0.003 | TATTOO - JEANS | 0.002 | 0.004 | 0.035 |
| HONG - KONG | 0.014 | 0.014 | 0.925 | THAT - ALSO | 0.301 | 0.166 | 0.376 |
| HOSTED - DRUGS | 0.016 | 0.013 | 0.013 | THIS - CITY | 0.423 | 0.123 | 0.132 |
| INTERVIEWS - FOURTH | 0.012 | 0.011 | 0.031 | THEIR - SUPPORT | 0.165 | 0.117 | 0.189 |
| KANSAS - PROPERTY | 0.017 | 0.045 | 0.052 | THEIR - VIEW | 0.165 | 0.103 | 0.151 |
| KIRIBATI - GAMBIA | 0.003 | 0.003 | 0.712 | THEM - OF | 0.112 | 0.570 | 0.187 |
| LAST - THIS | 0.135 | 0.423 | 0.221 | THEN - NEW | 0.104 | 0.291 | 0.192 |
| LEAST - ROMANCE | 0.046 | 0.007 | 0.019 | THINKS - LOT | 0.007 | 0.040 | 0.079 |
| LIME - REGISTERED | 0.002 | 0.030 | 0.004 | TIME - OUT | 0.189 | 0.191 | 0.366 |
| LINKS - TAKE | 0.191 | 0.105 | 0.134 | TIME - WELL | 0.189 | 0.120 | 0.299 |
| LINKS - THAN | 0.191 | 0.125 | 0.141 | TOP - AS | 0.140 | 0.280 | 0.217 |
| MAIL - AND | 0.160 | 0.554 | 0.192 | TOP - COPYRIGHT | 0.140 | 0.218 | 0.149 |
| MAIL - BACK | 0.160 | 0.160 | 0.132 | TOP - NEWS | 0.140 | 0.168 | 0.192 |
| MAKE - LIKE | 0.141 | 0.140 | 0.297 | UP - AND | 0.200 | 0.554 | 0.334 |
| MANAGING - LOCK | 0.010 | 0.006 | 0.010 | UP - HAS | 0.200 | 0.228 | 0.312 |
| MANY - US | 0.116 | 0.284 | 0.210 | US - BE | 0.284 | 0.323 | 0.335 |
| MASS - DREAM | 0.016 | 0.017 | 0.048 | VIEW - IN | 0.103 | 0.540 | 0.153 |
| MAY - HELP | 0.184 | 0.156 | 0.206 | VIEW - PEOPLE | 0.103 | 0.121 | 0.138 |
| MOST - HOME | 0.141 | 0.365 | 0.207 | WALKED - ANTIVIRUS | 0.006 | 0.002 | 0.002 |
| NAME - IN | 0.144 | 0.540 | 0.207 | WEB - GO | 0.194 | 0.111 | 0.138 |
| NEITHER - FIGURE | 0.011 | 0.016 | 0.085 | WELL - INFO | 0.120 | 0.138 | 0.110 |
| NET - SO | 0.101 | 0.154 | 0.112 | WELL - NEWS | 0.120 | 0.168 | 0.161 |
| NEW - PLEASE | 0.291 | 0.168 | 0.205 | WEEKS - LONDON | 0.028 | 0.032 | 0.050 |

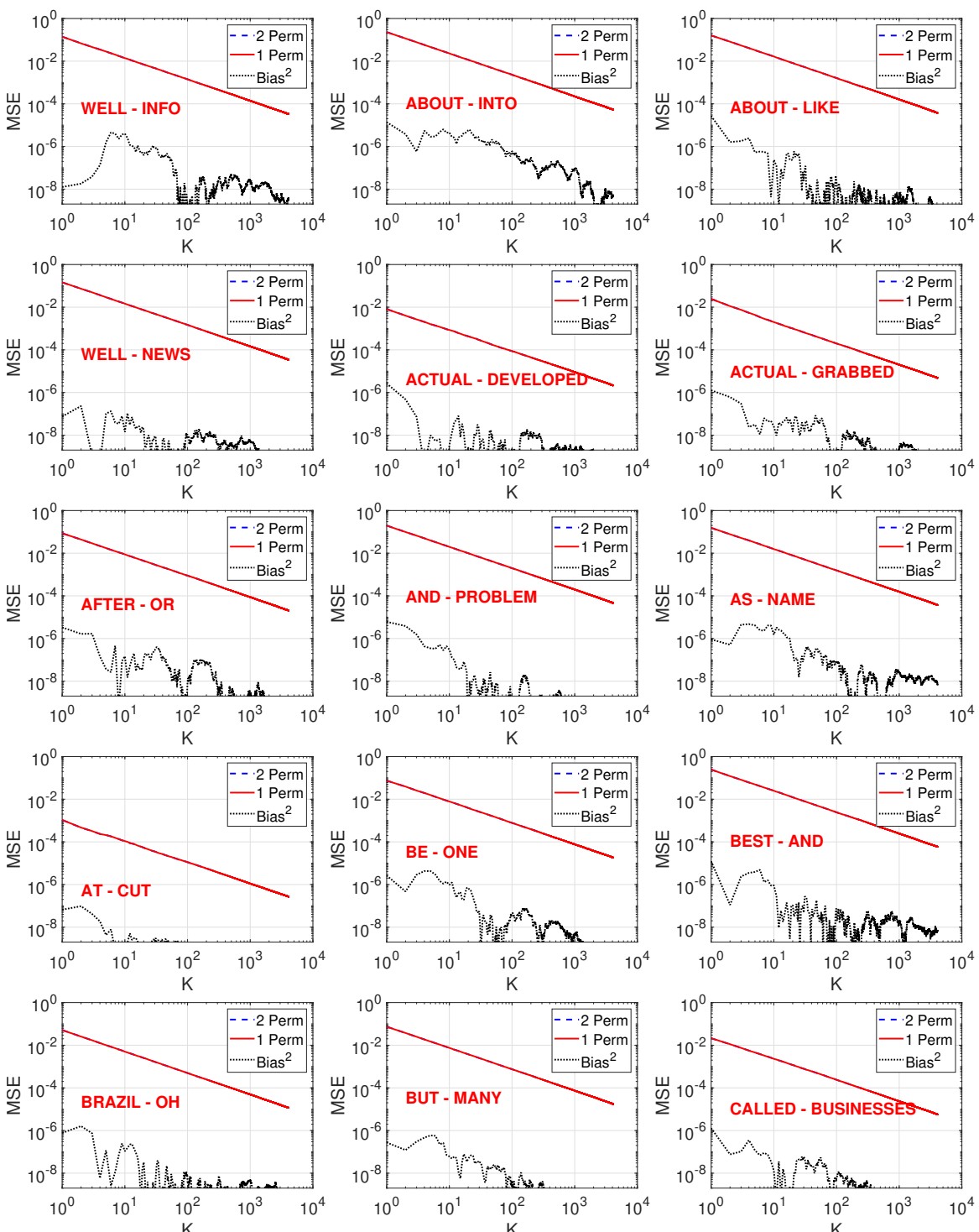

Figure 9: Empirical MSEs of C-MinHash-$(\pi, \pi)$ ("1 Perm", red, solid) vs. C-MinHash-$(\sigma, \pi)$ ("2 Perm", blue, dashed) on various data pairs from the *Words* dataset. We also report the empirical bias$^2$ for C-MinHash-$(\pi, \pi)$ to show that the bias is so small that it can be safely neglected. The empirical MSE curves for both estimators essentially overlap for all data pairs.

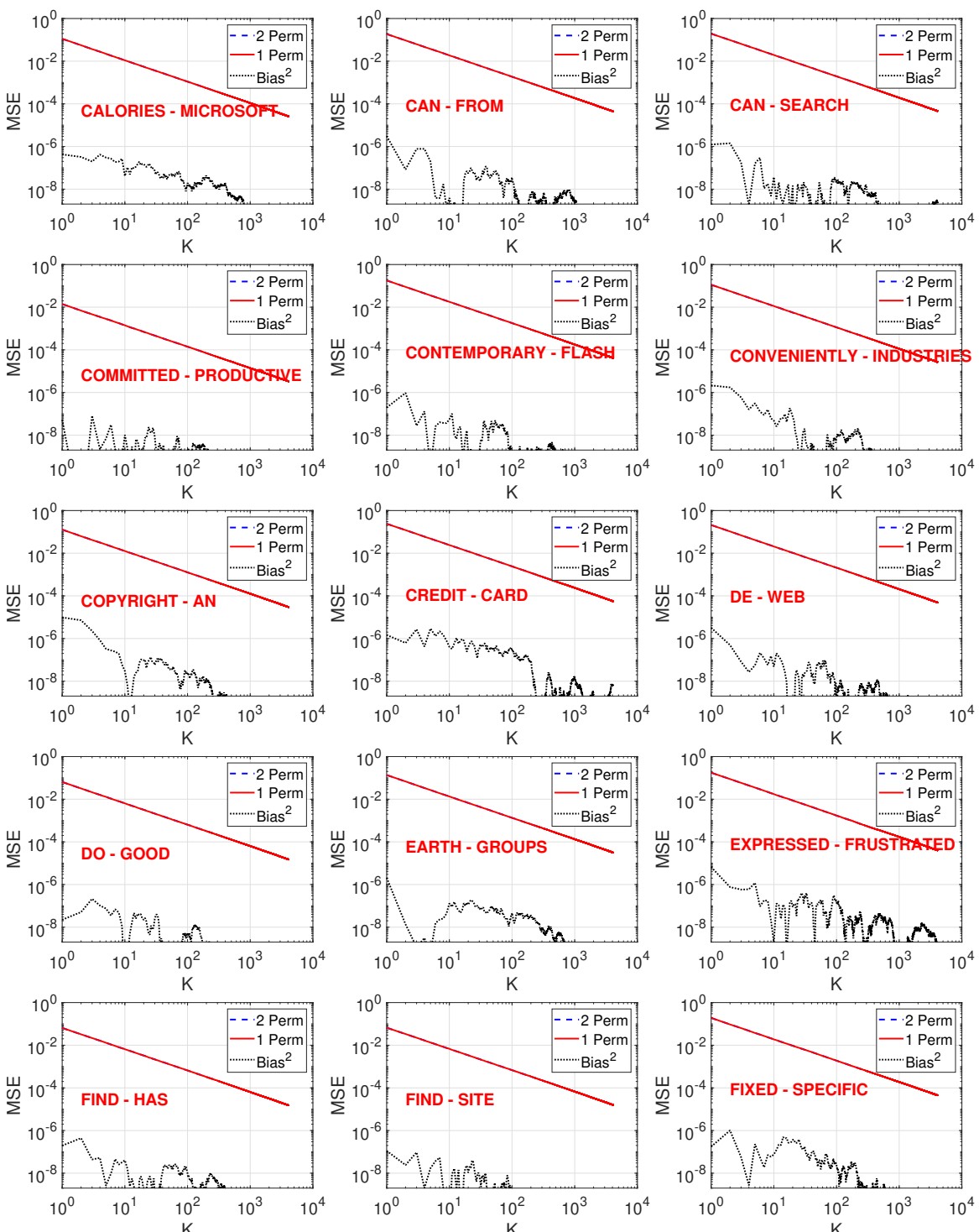

Figure 10: Empirical MSEs of C-MinHash-$(\pi, \pi)$ ("1 Perm", red, solid) vs. C-MinHash-$(\sigma, \pi)$ ("2 Perm", blue, dashed) on various data pairs from the *Words* dataset. We also report the empirical bias$^2$ for C-MinHash-$(\pi, \pi)$ to show that the bias is so small that it can be safely neglected. The empirical MSE curves for both estimators essentially overlap for all data pairs.

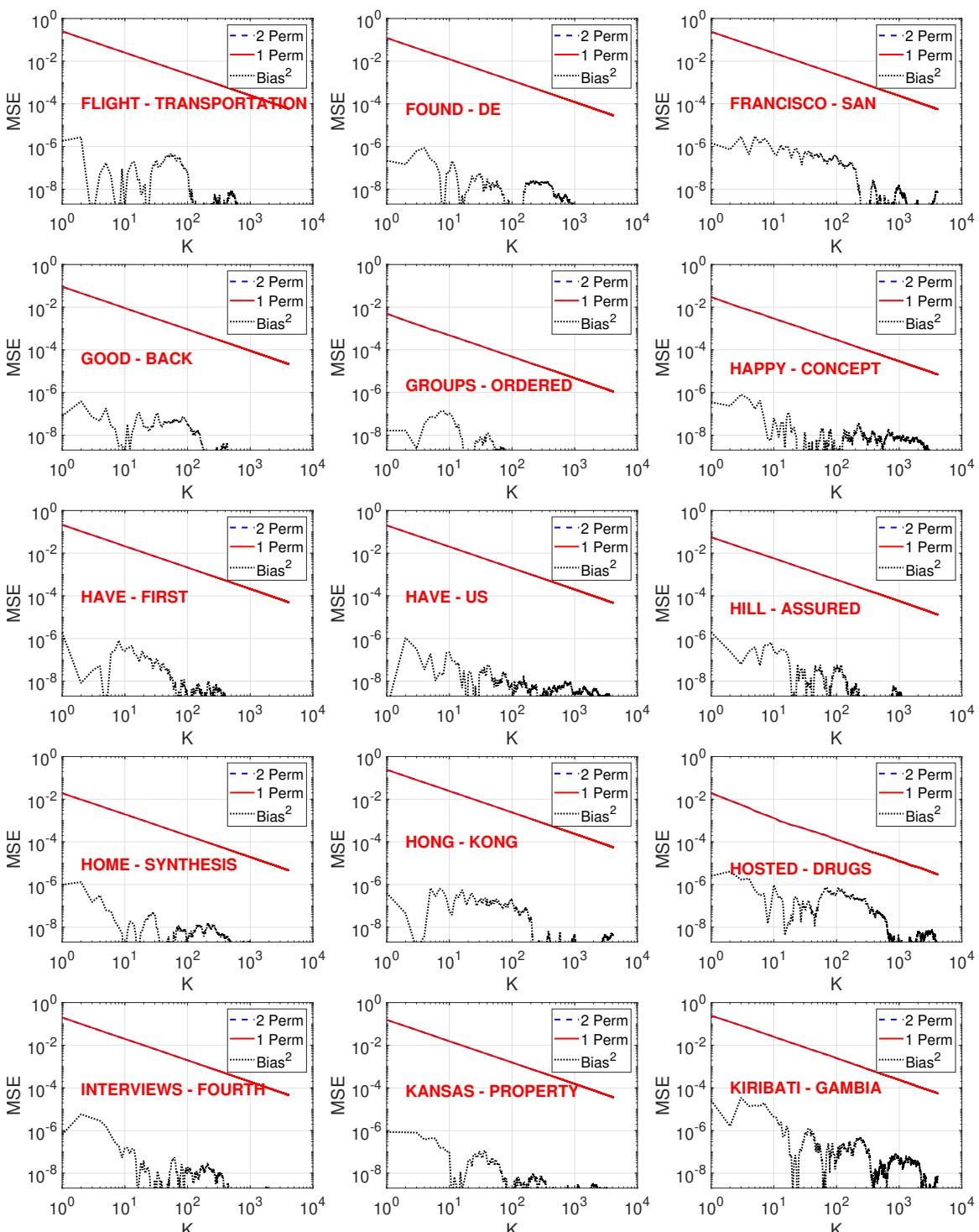

Figure 11: Empirical MSEs of C-MinHash-$(\pi, \pi)$ ("1 Perm", red, solid) vs. C-MinHash-$(\sigma, \pi)$ ("2 Perm", blue, dashed) on various data pairs from the *Words* dataset. We also report the empirical bias$^2$ for C-MinHash-$(\pi, \pi)$ to show that the bias is so small that it can be safely neglected. The empirical MSE curves for both estimators essentially overlap for all data pairs.

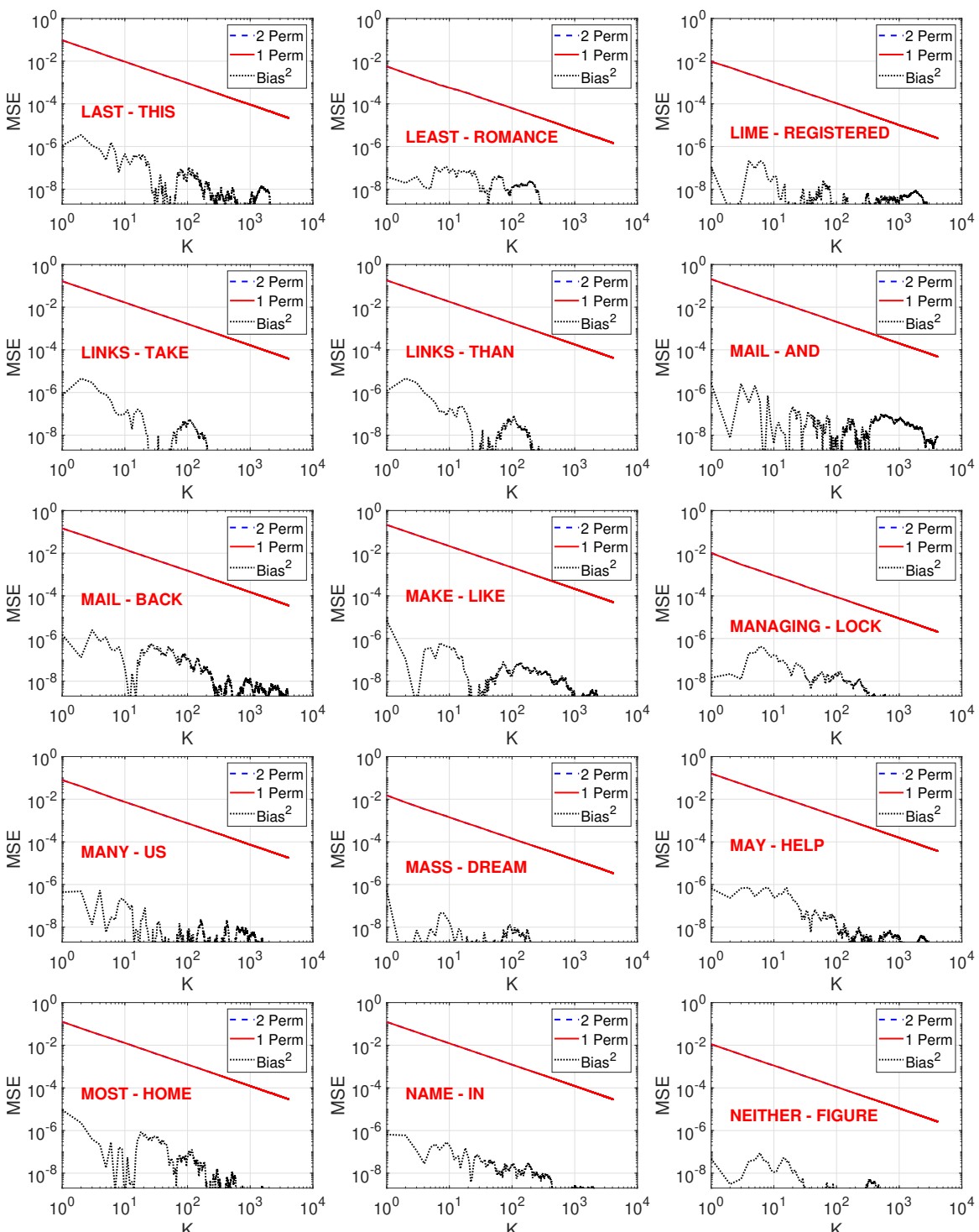

Figure 12: Empirical MSEs of C-MinHash-$(\pi, \pi)$ ("1 Perm", red, solid) vs. C-MinHash-$(\sigma, \pi)$ ("2 Perm", blue, dashed) on various data pairs from the *Words* dataset. We also report the empirical bias$^2$ for C-MinHash-$(\pi, \pi)$ to show that the bias is so small that it can be safely neglected. The empirical MSE curves for both estimators essentially overlap for all data pairs.

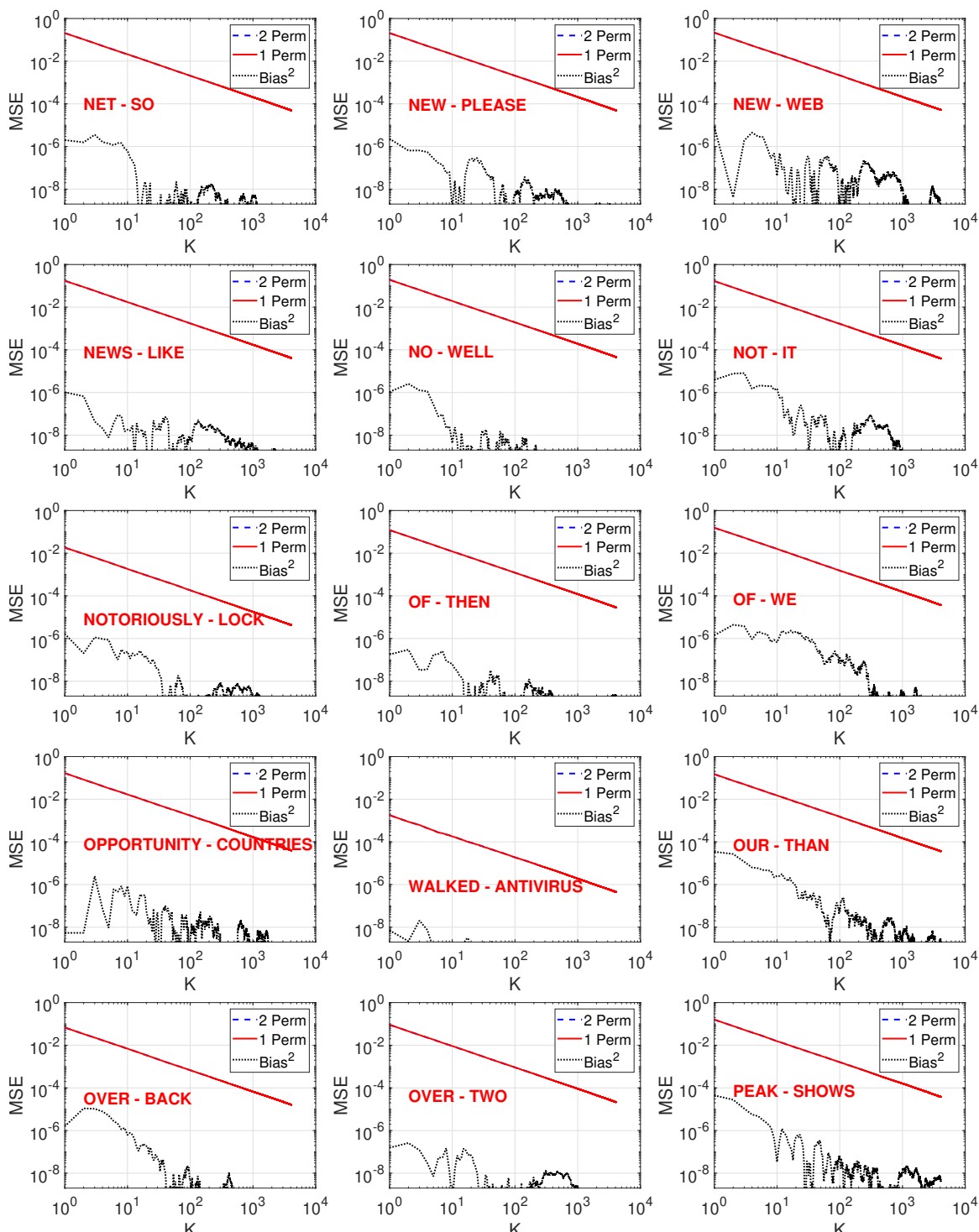

Figure 13: Empirical MSEs of C-MinHash-$(\pi, \pi)$ ("1 Perm", red, solid) vs. C-MinHash-$(\sigma, \pi)$ ("2 Perm", blue, dashed) on various data pairs from the *Words* dataset. We also report the empirical bias$^2$ for C-MinHash-$(\pi, \pi)$ to show that the bias is so small that it can be safely neglected. The empirical MSE curves for both estimators essentially overlap for all data pairs.

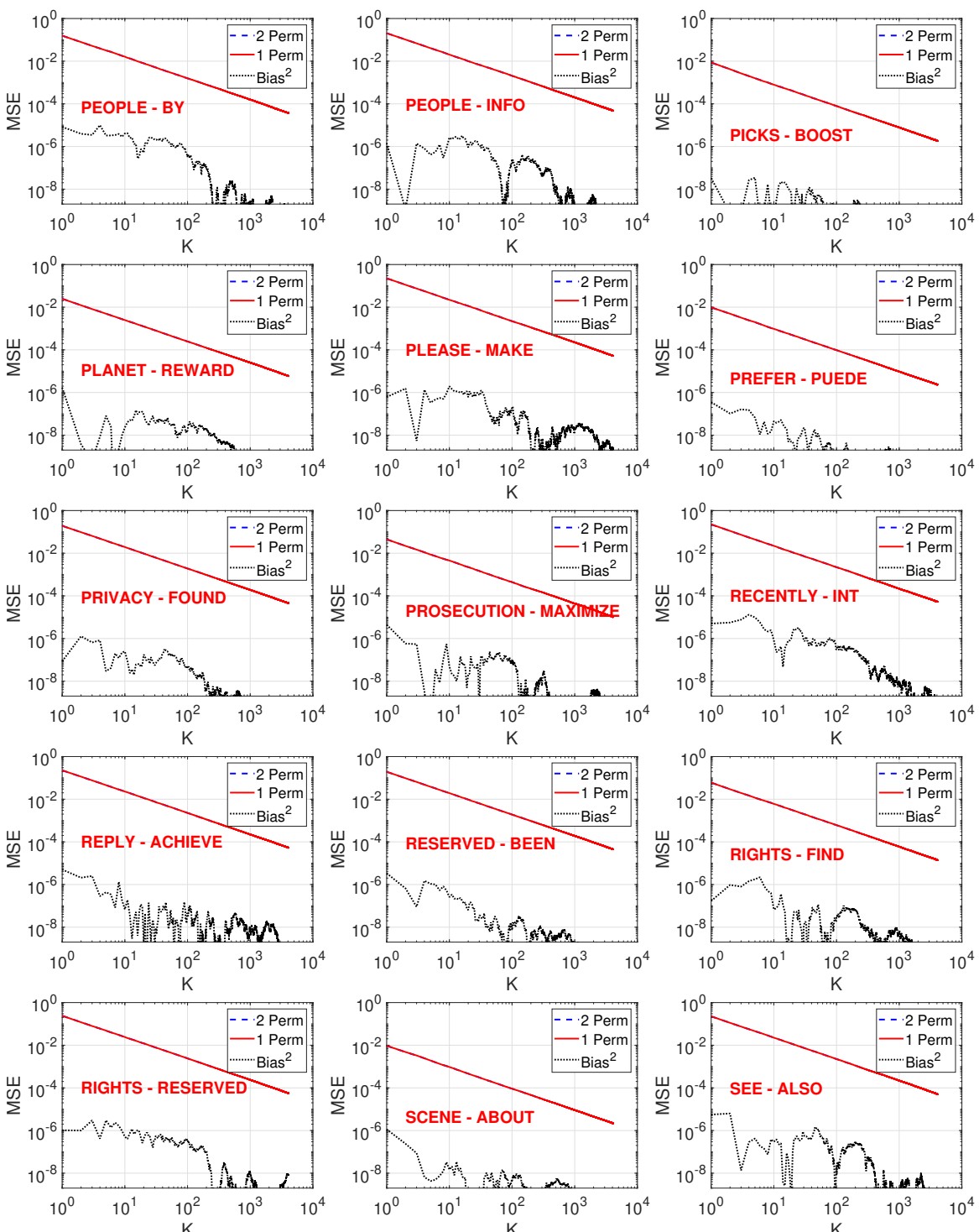

Figure 14: Empirical MSEs of C-MinHash-$(\pi, \pi)$ ("1 Perm", red, solid) vs. C-MinHash-$(\sigma, \pi)$ ("2 Perm", blue, dashed) on various data pairs from the *Words* dataset. We also report the empirical bias$^2$ for C-MinHash-$(\pi, \pi)$ to show that the bias is so small that it can be safely neglected. The empirical MSE curves for both estimators essentially overlap for all data pairs.

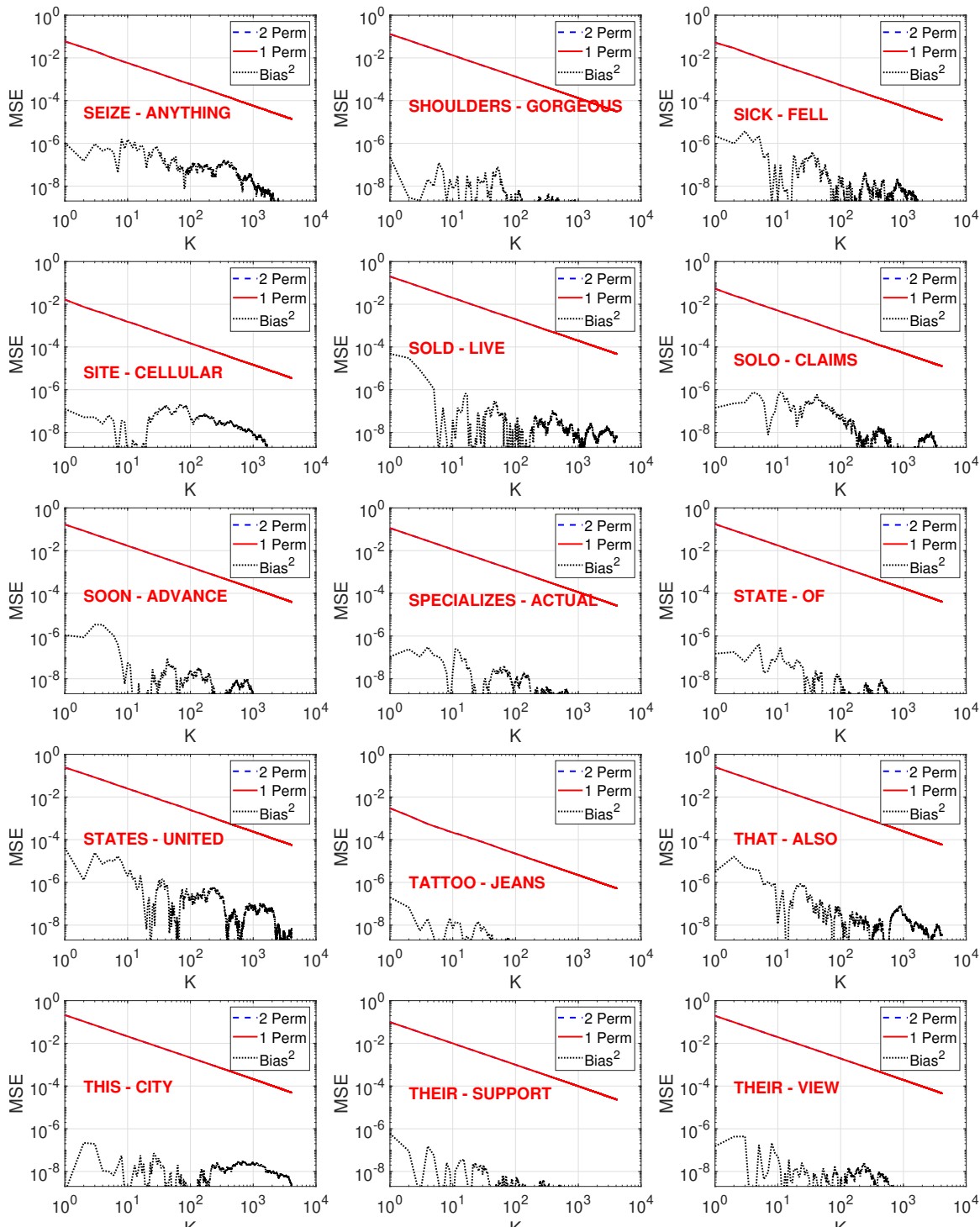

Figure 15: Empirical MSEs of C-MinHash-$(\pi, \pi)$ ("1 Perm", red, solid) vs. C-MinHash-$(\sigma, \pi)$ ("2 Perm", blue, dashed) on various data pairs from the *Words* dataset. We also report the empirical bias$^2$ for C-MinHash-$(\pi, \pi)$ to show that the bias is so small that it can be safely neglected. The empirical MSE curves for both estimators essentially overlap for all data pairs.

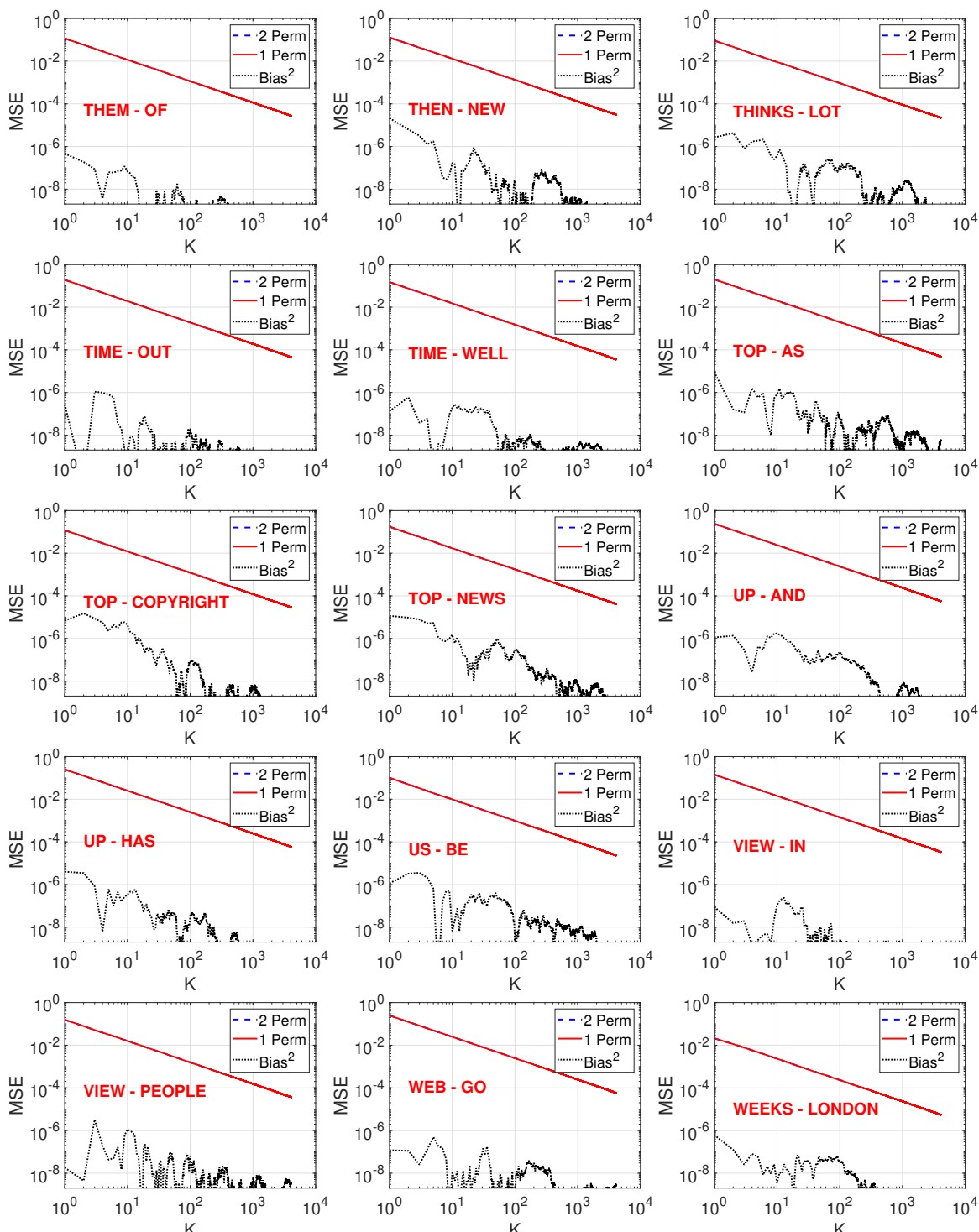

Figure 16: Empirical MSEs of C-MinHash-$(\pi, \pi)$ ("1 Perm", red, solid) vs. C-MinHash-$(\sigma, \pi)$ ("2 Perm", blue, dashed) on various data pairs from the *Words* dataset. We also report the empirical bias$^2$ for C-MinHash-$(\pi, \pi)$ to show that the bias is so small that it can be safely neglected. The empirical MSE curves for both estimators essentially overlap for all data pairs.

