# OpenReview forum: "C-MinHash: Improving Minwise Hashing with Circulant Permutation"
_ICLR.cc/2022/Conference — ICLR 2022 Submitted_

### Official Review · Reviewer_7874 · 2021-10-24

**Correctness:** 4
**Technical Novelty And Significance:** 4
**Empirical Novelty And Significance:** 2
**Recommendation:** 8
**Confidence:** 3

**Main Review:**

This is an interesting and surprising result as the algorithm is neat and simple, and improves on a classical algorithm. The proof seems straightforward with laborious calculations, which I did not check carefully. Hence, intuition as to why it works is important. Unfortunately, the paper does not provide the intuition at all and I hope the authors can elaborate on this.

From an algorithmic perspective, since the variance depends on the Jaccard similarity itself, one can only assume the worst case that the variance is 1/(4K) and pick K accordingly to obtain an (1+/-eps)-approximation. The new variance in Theorem 3.1 has an almost intractable non-closed expression for the variance. How does one choose K now? Is there a neater (though weaker) upper bound for the variance in Theorem 3.1?


**Summary Of The Paper:**

The paper designs an improved version of the classical MinHash data structure for calculating the Jaccard similarity between two binary strings.

The classical MinHash data structure generates K hash values for a binary string by generating K independent random permutations of the binary string and taking the index of the first “1” in the permuted string as the hash value in each random permutation. The estimator is then the percentage of equal hash values out of K hash values for the two binary strings. This is an unbiased estimator.

This paper’s algorithm, called C-MinHash, first applies a random permutation to the binary string. For the K hash values, instead of using K independent random permutations, the new algorithm uses the same permutation but shifted by 1, 2, …, K positions. The estimator is the same as that of the MinHash and can be easily shown to be unbiased. The paper then shows that this simple scheme, with only two independent random permutations, yields a smaller variance than that of MinHash.


**Summary Of The Review:**

Personally I like the paper and I hope that the paper will be accepted. ~However, I strongly feel that this paper should probably be merged with a separate paper by the same authors on reducing two permutations to one to tell a full story. I had a look at the arXiv version (https://arxiv.org/pdf/2109.04595.pdf) of the other paper and the main proof is just 4 pages. There is no compelling reason to split them into two papers. ~

---

> ### Author Response · Authors · 2021-11-19
> **Response to Reviewer 7874**
>
> Dear Reviewer,
>
> Thank you so much for the positive feedback and good suggestions.
>
> We fully agree that the results about "reducing two permutations to one" should be an integral part of the paper. Please kindly check Theorem 5.1 (on page 8) in the submission, which reports the theoretical expectation of the estimator after re-using the permutation. The proof of Theorem 5.1 can be found on page 24 - 27, which is exactly 4 pages coincidentally.  We realize that we should have emphasized Theorem 5.1 and related results a bit more in the introduction, abstract, conclusion, etc.  We appreciate your kind suggestion.
>
> Indeed, during the development of this work (which lasted for quite a few years), it was also a big surprise/shock to us when we initially obtained the theoretical results (after seeing the empirical evidence) that the estimation variance by using two permutations is strictly smaller than the variance of the classical minhash using K independent permutations. As in (7) at the bottom of page 4, the estimator is written as
>
> \begin{equation}
>     \hat J_{\sigma,\pi}=\frac{1}{K}\sum_{k=1}^K 1[h_k(v)=h_k( w)]
> \end{equation}
>
> While the above estimator has the same format as the classical MinHash, the difference is that the  indicators in (7)  $1[h_k(v)=h_k( w)]$ are correlated, not independent. Obviously, as long as the the indicators are "pairwise negatively correlated", then the variance of the new estimator would be smaller than that of the classical MinHash. Our Theorem 3.1 and Theorem 3.4 basically proved this "pairwise negatively correlated" property (among other results):
>
> $Cov(1[h_i(v)=h_i(w)], 1[h_j(v)=h_j(w)])<0$, $\forall i\neq j$.
>
> We would be happy to write the above expression explicitly after (7), before presenting  Theorem 3.1 and Theorem 3.4.  We hope this provides more intuition which might help readers understand the fact on variance reduction.
>
> The variance of classical MinHash is $J(1-J)/K$, which is an upper bound of our variance and also contains the "$J$" to be estimated. In a sense, choosing $K$ for classical MinHash faces the same issue as for our new estimator. This is not a big issue in practice, because one can often choose $K$ based on the target $J$ and some reasonable criterion such as coefficient of variation.  The good thing is that for any chosen $K$, the new estimator always enjoys strictly smaller variance than the classical MinHash.
>
> Again, we highly appreciate your positive feedback and kind suggestions.

---

> > ### Comment · Reviewer_7874 · 2021-11-20
> > **Thanks for the response**
> >
> > Thanks for pointing out that the summands are negatively correlated. This should definitely be made more explicit in the paper.
> >
> > Thanks for pointing me to Section 5. I'm sorry that I overlooked it earlier.

---

> > > ### Author Response · Authors · 2021-11-21
> > > **Thank you for your suggestion**
> > >
> > > Dear Reviewer:
> > >
> > > Thank you again for your nice suggestion of adding the "pairwise negatively correlated" result in the main paper. We have added Proposition 3.1 on top of page 5, right after the definition of the estimator (7), to provide an immediate intuition about the source of variance reduction. To further assist some readers, we also provide the detailed step-by-step elementary proof of Proposition 3.1.  By using Theorem 3.5 (original Theorem 3.4), the proof of Proposition 3.1 is essentially trivial but we hope those elementary steps would help readers better understand our theoretical results as well as the basics of variance calculations. Note that the proof of Proposition 3.1 does not  really need Theorem 3.5, but then it would be essentially the same proof as the proof for Theorem 3.5.
> > >
> > > We hope this treatment, i.e., adding Proposition 3.1 just for providing the intuition (and introductions of basic statistics) without modifying other parts of the paper, might be a good solution to address your kind suggestion. Thank you.

---

### Official Review · Reviewer_5hPp · 2021-10-29

**Correctness:** 4
**Technical Novelty And Significance:** 3
**Empirical Novelty And Significance:** 2
**Recommendation:** 5
**Confidence:** 4

**Main Review:**

Basically, I like the paper's motivation; it is quite a fundamental research problem to approximate the Jaccard similarity for large-scale data efficiently. Besides, this paper is well structured, and the theoretical background of the proposed approach is well described in the paper. Specifically, I really appreciate that this paper conducted an excellent job revealing the proposed approach's theoretical approximation quality. Furthermore, it experimentally confirms that the experimental results follow the theoretical results.

However, I am concerned about the experiment since the paper compares the proposed approach to only the original approach of MinHash. Since MinHash is a popular approach with a long history, many variant approaches have been proposed below. Therefore, it needs to compare the proposed approach to the previous approach to demonstrate its usefulness.

- Li et al., b-Bit Minwise Hashing for Estimating Three-Way Similarities
- Ioffe, Improved Consistent Sampling, Weighted Minhash and L1 Sketching
- Manasse et al., Consistent weighted sampling
- Shrivastava, Exact Weighted Minwise Hashing in Constant Time

**Summary Of The Paper:**

This paper proposes an effective approach for MinHash by permutating data vectors. It first randomly shuffles the data to break structures exhibited in the original data and then performs permutation K-times to obtain K hash values. Besides, this paper proposes an approach that performs only one permutation to compute hash values. This paper shows the theoretical approximation error of the proposed approach. By using text and image datasets, it shows that experimental results follow the results of the theoretical analysis.

**Summary Of The Review:**

This paper is well-written.
It needs to compare the proposed approach to the previous approaches.

---

> ### Author Response · Authors · 2021-11-19
> **Response to Reviewer 5hPp**
>
> Dear Referee:
>
> Thank you so much for the very generous nice comments on our submission; your feedback is highly appreciated.
>
> Thanks for providing the additional references. The mentioned References [2-4]:
>
> [2] Ioffe, Improved Consistent Sampling, Weighted Minhash and L1 Sketching
>
> [3] Manasse et al., Consistent weighted sampling
>
> [4] Shrivastava, Exact Weighted Minwise Hashing in Constant Time
>
> are all about consistent weighted sampling (CWS), which is exactly the same as minwise hashing (MinHash) when the data are binary (0/1). Therefore, comparing with [2-4] on binary data is the same as comparing with the original MinHash.
>
> The mentioned Reference [1]
>
> [1] Li et al., b-Bit Minwise Hashing for Estimating Three-Way Similarities
>
> published in NIPS'10, was about using the standard (b-bit) minwise hashing for estimating three-way similarity as the title suggested.
>
> The work in our submission is a surprising and fundamental result, as Reviewers correctly pointed out, for the classical MinHash developed over 20 years ago. Therefore, our work will serve as the basic building block for other hashing methods based on minwise hashing. One such example is "One Permutation Hashing (OPH)" as mentioned at the bottom of page 2 in the submission. As we have explained in the general response, using circulant permutations to improve OPH is a fairly straightforward application. Since this submission is already quite long (37 pages) and self-contained, we decided to report the use of C-MinHash to improve OPH in a separate manuscript which we promised to share once requested. Therefore, during this rebuttal, we provide the link to the mentioned manuscript on "Circulant OPH (C-OPH)".
>
> https://anonymous.4open.science/r/report_COPH-F4BC/Circular_OPH_anonymous_report.pdf
>
> In summary, while we feel this work is self-contained and complete as it is,  we will be more than happy to add references to CWS, three-way similarities, OPH, etc. We ourselves are also extremely familiar with those lines of researches. Thank you.

---

### Official Review · Reviewer_89pE · 2021-11-02

**Correctness:** 4
**Technical Novelty And Significance:** 3
**Empirical Novelty And Significance:** 2
**Recommendation:** 5
**Confidence:** 4

**Main Review:**

The fact that using two permutations provides smaller than vanilla MINHASH is interesting, and the theoretical analysis of the paper is extensive. I think the paper can improve in the following perspective.
1. There are some works on improving MINHASH, such as one permutation hashing and densifying one permutation hashing. The relation between MINHASH and these works needs to be thoroughly discussed. Instead, the paper only says C-MINHASH can be used to improve one permutation hashing. Moreover, experiments should also be conducted to compare with these works.
2. The benefits of using only two permutation is not well justified. Existing works (one permutation hashing and densifying one permutation hashing) only need to conduct permutation once while MINHASH still needs to conduct permutation K times. The storage cost may not be a concern as one can generate random permutations on the fly without storing them.
I will raise my score if the concerns are addressed.


**Summary Of The Paper:**

This paper proposes C-MINHASH to improve vanilla MINHASH. Instead of using K random permutations to generate K hash values, C-MINHASH requires only two permutations. Theoretically, C-MINHASH provides unbiased estimate, and its variance is smaller than MINHASH. Extensive empirical experiments verify the theoretical analysis.

**Summary Of The Review:**

The method and theoretical analysis are interesting. However, the paper lacks a clear justification of the benefits of C-MINHASH and a through comparison with related works.

---

> ### Author Response · Authors · 2021-11-19
> **Response to Reviewer 89pE**
>
> Dear Reviewer,
>
> We highly appreciate your valuable comments.  Regarding the relationship with One Permutation Hashing (OPH), please kindly check our general response and the anonymized report
>
> https://anonymous.4open.science/r/report_COPH-F4BC/Circular_OPH_anonymous_report.pdf
>
> which shows how we could incorporate the ideas of circulant permutations to improve OPH and obtain the C-OPH (Circulant OPH) algorithm. As mentioned in this submission (bottom of page 2) , that report was an application of C-MinHash and would be regarded as a separate work. The basic idea C-OPH is  simple: we utilize one shorter permutation in a circulant shifting manner in all bins. Using the circulant ideas of C-MinHash,  C-OPH improves the accuracy of densified OPH, at the same computational cost. A nice consequence of C-OPH is that it effectively only needs 1/K permutation instead of one permutation.  We feel that C-MinHash should not be regarded as an competent with OPH, rather, it is a building block to improve other algorithms including OPH.
>
> Therefore, we hope it is clear that this current submission (on C-MinHash) has provided the necessary fundamental understanding and theory of the circulation trick. The submission (which is 37-page long) is by itself a complete work with self-contained motivation, analysis and results.  C-OPH uses C-MinHash and requires independent detailed introduction and efforts, and is hence more suitable for a separate manuscript.
>
> We hope our explanations and the provided anonymized report on C-OPH would be able to address your concerns. Please kindly let us know if we could supply more details to assist you better understand/appreciate our contributions. Thank you.

---

### Official Review · Reviewer_yZFL · 2021-11-02

**Correctness:** 3
**Technical Novelty And Significance:** 3
**Empirical Novelty And Significance:** 3
**Recommendation:** 6
**Confidence:** 4

**Main Review:**

Pros:
1. I like the idea of using circulation and shifting for the hashing strings, which can reduce the number of permutations but still introduce randomness (even though not entirely independent).

2. The authors systematically show that C-MinHash-($\sigma$,$\pi$) has uniformly lower Jaccard estimation variance than the vanilla MinHash.

3. The one permutation variant C-MinHash-($\pi$,$\pi$) is practically interesting, which can use a single permutation to achieve very similar results as C-MinHash-($\sigma$,$\pi$).

4. Extensive experiments justify their theoretical analysis over MinHash.

Cons:
1. As claimed in the conclusion section, there exist other works improving MinHash with much fewer permutations, such as [1,2,4,6]. Thus, it will be more convincing to show more baselines in the numerical experiments in Section 4 rather than using MinHash only.

2. Many important related works about MinHash are missing, such as [1-6]. I suggest the authors discuss more related works and illustrate their difference to C-MinHash.

Reference:

[1] Li, Ping, Art Owen, and Cun-hui Zhang. "One Permutation Hashing." Advances in Neural Information Processing Systems 25 (2012): 3113-3121.

[2] Shrivastava, Anshumali, and Ping Li. "Densifying one permutation hashing via rotation for fast near neighbor search." In International Conference on Machine Learning, pp. 557-565. PMLR, 2014.

[3] Shrivastava, Anshumali, and Ping Li. "Improved densification of one permutation hashing." In Proceedings of the Thirtieth Conference on Uncertainty in Artificial Intelligence, pp. 732-741. 2014.

[4] Shrivastava, Anshumali. "Optimal densification for fast and accurate minwise hashing." In International Conference on Machine Learning, pp. 3154-3163. PMLR, 2017.

[5] Li, Ping, Xiaoyun Li, and Cun Hui Zhang. "Re-randomized densification for one permutation hashing and bin-wise consistent weighted sampling." Advances in Neural Information Processing Systems 32 (2019).

[6] Jia, Peng, Pinghui Wang, Junzhou Zhao, Shuo Zhang, Yiyan Qi, Min Hu, Chao Deng, and Xiaohong Guan. "Bidirectionally Densifying LSH Sketches with Empty Bins." In Proceedings of the 2021 International Conference on Management of Data, pp. 830-842. 2021.

**Summary Of The Paper:**

Min-wise hashing (MinHash) is a fundamental and popular algorithm in machine learning. This paper proposes Circulant MinHash (C-MinHash) to approximate the Jaccard similarity in massive binary data. Compared with MinHash, C-MinHash only requires two (or maybe one in practice) random permutations in a circulant manner for approximation. The authors also systematically demonstrate that the C-MinHash can provide a smaller estimation variance than MinHash. Extensive experiments validate the effectiveness of C-MinHash.



**Summary Of The Review:**

Overall, the idea of this paper looks very promising, and the authors also achieve better theoretical results in terms of lower Jaccard estimation variance. However, the experiments and the related work are a lit weak. Such limitations justify my initial rating.

---

> ### Author Response · Authors · 2021-11-19
> **Response to Reviewer yZFL**
>
> Dear Reviewer:
>
> Thanks for your valuable feedback and recognition of our contributions.
>
> As stated in the general response to all Referees and Area Chairs, C-MinHash is a basic piece of building block to improve the general minwise hashing-type algorithms, including OPH. Our anonymous report provided in the general response
>
> https://anonymous.4open.science/r/report_COPH-F4BC/Circular_OPH_anonymous_report.pdf
>
> shows how to use the circulation trick presented in this submission to improve the variance of the densified OPH [Li et al., 2019].
>
> As mentioned in the general response, we feel that C-MinHash should not be regarded as an competent with OPH, it is rather a building block to improve other algorithms including OPH. This submission (which is 37-page long) is by itself a complete work with self-contained motivation, analysis and results.
>
> Also, we would like to specifically thank you for providing the list of references "[1-6]". Incidentally, when we checked the anonymized report, those papers were all included in the bibliography. In particular, the nice idea in "[6]" by Jia Peng, Pinghui Wang, etc (SIGMOD 2021) could be used to improve the processing efficiency of any densification method with the same estimation accuracy (which means "[6]" can be combined with C-OPH too).
>
> We hope our response and anonymized  report on C-OPH (as an example of applications of C-MinHash) could adequately address your concern.

---

> > ### Comment · Reviewer_yZFL · 2021-11-21
> > **Feedback to the authors**
> >
> > Thank you for your response and the anonymous report.
> >
> > Based on the anonymous report and its experiments, the authors show that the circulant permutation idea can also be used to OPH [5] and improve its accuracy (with lower MSE). Thus, my first concern about the performance has been addressed.
> >
> > The anonymous report also discussed the related work about the OPH and its variants (in sections 1.2 and 2). If the authors add this anonymous report into the supplementary, I think my second concern can be addressed. However, this anonymous report is separated and unknown to other readers. If those readers only read the manuscript and the supplementary, I do not think this work is complete because the related work is still missing.
> >
> > Well, basically, I agree with the authors that the circulant permutation idea is a building brick that can improve the accuracy of MinHash and many of its variants. However, separating this idea into two articles makes both of them weak. As they share the same philosophy, I suggest the authors incorporate them into one paper; that will be more convincing and address most concerns to me and other reviewers.
> >
> > As the authors address my concerns about the experiments, I am glad to increase my rating to 6. However, since I still have concerns about the related work, I cannot champion this paper. Thank you.

---

> > > ### Author Response · Authors · 2021-11-24
> > > **Thanks for your support and raising the score**
> > >
> > > Dear Reviewer,
> > >
> > > We highly appreciate your support and acknowledging that the anonymous report has addressed your concerns. It is also nice of you to agree that "the circulant permutation idea is the building brick that can improve the accuracy of MinHash and many of its variants".
> > >
> > > Indeed, these are two mechanisms to improve MinHash: (1) using binning as in OPH; and (2) using circulant permutations as in this submission.  Circulant ideas can actually also improve other related algorithms in addition to densified OPH.
> > >
> > > We will be more than happy to add OPH references to [1]-[6] in the main paper. We will also be happy to add references to CWS algorithms as other Reviewer 5hPp suggested.
> > >
> > > Please be assured that the anonymous report will eventually be made available in the public domain. It is not that we hope to publish one more paper. Again, as this paper is already quite long (37 pages) and in many ways self-contained, appending a separate report, which is related but also needs its own introduction, motivation, and notation, would require some non-trivial effort  not only from authors but also from readers. We will think about this carefully.
> > >
> > > Thanks again for your support and feedback.

---

### Author Response · Authors · 2021-11-19
**General Response about the connection to OPH**

Dear Referees and Area Chairs,

We sincerely appreciate your effort in reviewing our submission and all the constructive feedback. It seems that all the reviewers agree that:
1) our findings on the variance reduction of using circulant permutations in MinHash is interesting and fundamental;
2) our theoretical results are correct and backed up by extensive experiments.

The main question asked by the reviewers is the comparison of our C-MinHash with One Permutation Hashing (OPH) [Li et al., NIPS 2012] and its related densification methods.

In the original submission, we mentioned at the bottom of page 2 (as well as the Conclusion) that ''... the proposed C-MinHash mechanism can also be conveniently used as a tool to improve more MinHash based algorithms,  for example, the One Permutation Hashing ... Since such combination is a research direction which requires independent introduction and efforts, we present the work in a separate manuscript (which could be anonymously shared with Referees if needed). ''

Since Referees indeed requested to read the separate manuscript, we've made the mentioned report on Circulant OPH (C-OPH) anonymous and would like to share the report  with Referees and Area Chair:
https://anonymous.4open.science/r/report_COPH-F4BC/Circular_OPH_anonymous_report.pdf

The manuscript  on C-OPH shows that  1) C-MinHash can  be used as a building block to improve the accuracy of One Permutation Hashing (OPH) and its best (most accurate) densification method. 2) effectively, C-OPH just needs $1/K$ permutation instead of one permutation.

The idea of C-OPH is a fairly straightforward application of C-MinHash. That is, after the data are broken into $K$ bins, we use a smaller permutation of length $D/K$ (instead of $D$) to generate hashes in a circulant fashion.

Therefore, we feel C-MinHash should not be regarded as an competent with OPH, it is rather a building block to improve other algorithms including OPH. This submission (which is 37-page long) is by itself a complete work with self-contained motivation, analysis and results.  C-OPH uses C-MinHash and requires independent detailed introduction and efforts, and is hence more suitable for a separate manuscript.

We hope it is clear that this current submission (on C-MinHash) has provided the necessary fundamental understanding and theory of the circulation trick. Again, we would like to sincerely thank all Referees and Area Chairs for their efforts to provide valuable comments on our submission.  We also provide a separate response to each individual Reviewer regarding their more specific questions.

---

> ### Author Response · Authors · 2021-11-21
> **Added Proposition 3.1 "pairwise negatively correlated" property**
>
> Dear Reviewers and Area Chair,
>
> Per the suggestion of Reviewer 7874, we have added Proposition 3.1 (on top of page 5) to emphasize that the indicators of estimator (7) are "pairwise negatively correlated" which provides the immediate intuition to help readers understand the source of variance reduction, i.e., Theorem 3.5 in the revised version.
>
> Note that the proof of Proposition 3.1 does not really need Theorem 3.5. However, using Theorem 3.5 leads to a very elementary proof which we expect some readers might appreciate, for example, the expansions of sum of random indicators as in text books.
>
> Again, thank you all very much for the great efforts in reviewing our work and providing constructive comments. Please let us know if there are further questions which we could help clarify.
>
> Sincerely,
>
> Authors

---

### Decision · Program_Chairs · 2022-01-20

**Decision:**

Reject

**Comment:**

This was a somewhat unusual submission in that the authors tried to motivate their paper by pointing to a separate anonymous manuscript.  However, the authors didn't seem to want to confirm they would merge the manuscripts when asked about this. It was thought that in fairness the submitted manuscript should be judged on its own. After discussion, it was agreed that the submitted paper on its own, did not generate enough enthusiasm to merit acceptance.